# Unlearning Diffusion Policies with Relative Fisher Forgetting

## Abstract

Diffusion policies have advanced offline reinforcement learning (RL) by enabling expressive, multi-modal action generation. As these models move closer to real deployment, it becomes critical to remove the influence of specific data whether for privacy compliance, to eliminate unsafe behaviors, or to satisfy regulatory requirements. Existing unlearning methods, however, are ineffective for diffusion policies, as training influence is dispersed across the denoising process and reinforced by critic values. We introduce Relative Fisher Forgetting (RFF), the first framework for unlearning in diffusion-based offline RL. RFF combines two complementary components: actor unlearning via noise-aware influence gradients scaled by relative Fisher importance, and critic unlearning through value suppression of unlearn data. To stabilize training, RFF alternates actor–critic updates and employs gradient clipping, retain-set regularization, and convergence monitoring. Experiments on MuJoCo benchmarks demonstrate that RFF reliably removes unlearn trajectories and behaviors while preserving performance on retained data, outperforming prior unlearning baselines in both efficacy and efficiency.

## 1 Introduction

Offline reinforcement learning (RL) (Kostrikov et al., 2021; Levine et al., 2020) offers a promising path for deploying learning-based decision-making systems in real-world domains where active environment interaction is costly, risky, or infeasible, such as autonomous driving (Fang et al., 2022), healthcare (Tang et al., 2022), and robotics (Sinha et al., 2022). Recently, a major trend has emerged around leveraging diffusion models originally designed for vision and language generation, as expressive policy classes in offline RL (Mao et al., 2024; Wang et al., 2022; Fang et al., 2024; Chi et al., 2023; Janner et al., 2022). Compared to traditional offline RL, diffusion-based approaches offer distinct advantages: they naturally support multi-modal action distributions and enable flexible, high-quality policy generation through iterative denoising. These properties make them particularly well-suited for capturing the complexity and uncertainty of real-world decision-making.

However, as offline RL systems advance to practical deployment, a critical problem emerges: machine unlearning (Nguyen et al., 2022; Bourtoule et al., 2021). Machine unlearning in offline RL aims to remove the influence of designated training data without the prohibitive cost of retraining. This capability is increasingly important for compliance with governance requirements (*e.g.*, the GDPR "right to be forgotten"), and for maintaining adaptive, trustworthy decision-making systems. In practice, unlearning may require removing specific trajectories or suppressing broader behavioral modes (*e.g.*, forgetting reckless driving), which is essential for safeguarding user privacy, eliminating unsafe or outdated behaviors, and enabling post-hoc correction or auditing of deployed policies.

**Unique Challenges.** While unlearning has been extensively studied in supervised learning (Bourtoule et al., 2021), and more recently in RL via trajectory or environment forgetting (Gong et al., 2024; Ye et al., 2023), it remains unexplored for diffusion-based offline RL, where policies are parameterized as conditional denoising processes. This gap is significant, as diffusion models are rapidly gaining traction in offline RL for their ability to represent complex action distributions and generate high-quality behaviors. Without effective unlearning mechanisms, these powerful models risk becoming rigid and non-compliant in practical deployments where data removal, correction, or adaptation is required. Yet, the generative nature of diffusion-based policies introduces unique challenges. *First,* actions in diffusion policies are produced via multi-step stochastic denoising, which diffuses

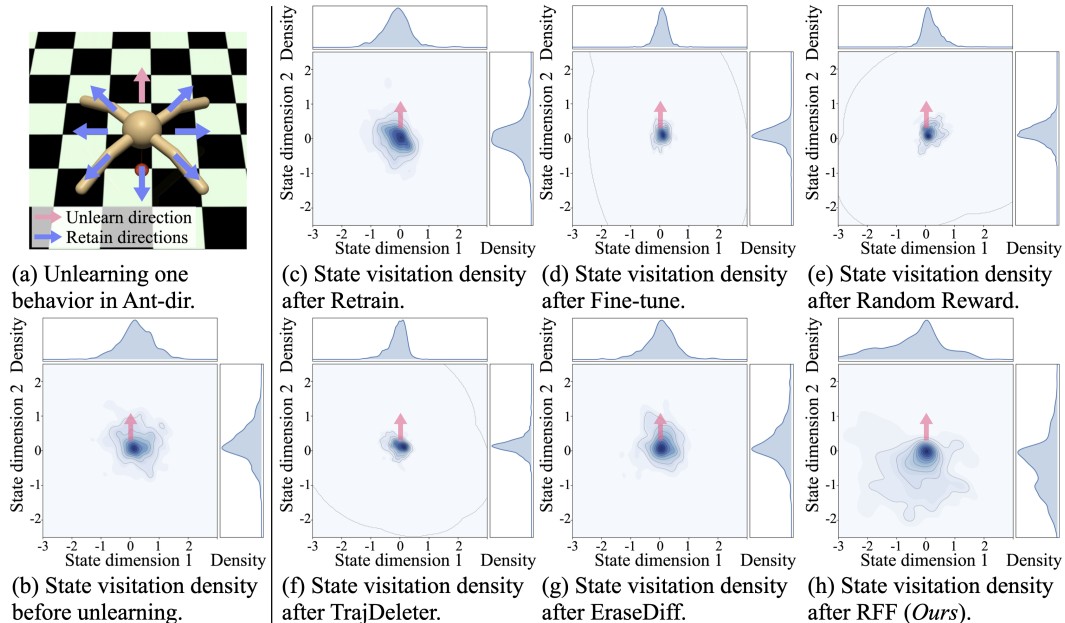

Figure 1: State visitation densities in Ant-dir. Darker regions indicate higher visitation frequency. Our RFF effectively suppresses the unlearn direction while preserving retain directions.

(a) Unlearning one behavior in Ant-dir.
(b) State visitation density before unlearning.
(c) State visitation density after Retrain.
(d) State visitation density after Fine-tune.
(e) State visitation density after Random Reward.
(f) State visitation density after TrajDeleter.
(g) State visitation density after EraseDiff.
(h) State visitation density after RFF (*Ours*).

the influence of individual training samples across the full sampling trajectory, making targeted removal difficult. *Second,* diffusion policies are trained using noise-conditioned reconstruction losses and reward supervision, complicating credit assignment and undermining reward-based unlearning strategies used in RL. *Third,* unlike supervised settings where data is *i.i.d.* and label-guided, offline RL data is temporally extended and governed by sequential transition dynamics, requiring unlearning methods to preserve long-horizon behavior and decision-making consistency.

**Limitations of State-of-the-Art.** These challenges render existing unlearning approaches inapplicable. For instance, methods like TrajDeleter (Gong et al., 2024) assume explicit, differentiable state-action policies and rely on $Q$-value suppression and policy training. They cannot be directly applied to denoising-based generative policies. Similarly, unlearning techniques developed for diffusion models, such as score function editing (Gandikota et al., 2023; 2024), knowledge distillation with filtered datasets(Zhou et al., 2025), or negative gradient updates (Wu et al., 2024; 2022), target vision or language generation, and lack support for reward-driven decision-making, sequential consistency, and offline RL constraints. As a result, unlearning in diffusion-based offline RL requires a new framework that aligns with both the denoising dynamics of diffusion models and the RL objective.

**Our RFF.** We propose Relative Fisher Forgetting (RFF), the first principled framework for unlearning in diffusion-based offline RL. RFF enables both trajectory- and behavior-level unlearning without retraining, while preserving utility on retained data and respecting the generative nature of diffusion policies. It introduces a dual-component scheme that jointly updates the actor and critic: the actor employs noise-aware gradients scaled by relative Fisher importance to attenuate the influence of unlearning data, while the critic applies value suppression to eliminate residual reward incentives tied to forgotten trajectories. By alternating actor and critic updates, RFF enforces consistency between action generation and value estimation, ensuring stable and effective unlearning under offline RL constraints. As illustrated in Fig. 1, RFF successfully suppresses the forgotten direction while preserving other valid behaviors. *Our contributions are as follows*[1]:

- We propose RFF, a general framework for diffusion-based offline RL that supports unlearning at both the trajectory-level (removing specific trajectories) and the behavior-level (suppressing policy modes induced by groups of trajectories).
- We develop a dual-component approach that combines *actor unlearning* via noise-aware, Fisher-weighted forgetting gradients with *critic unlearning* via value suppression, ensuring consistent removal of forgotten data across both policy and value functions.

---

[1]Our code is available at https://anonymous.4open.science/r/RFF-6720/.

- We conduct extensive empirical validation on MuJoCo single-task and multi-task benchmarks, showing that RFF achieves effective forgetting with minimal utility loss, outperforming retraining and approximate unlearning baselines in both fidelity and efficiency.

## 2 PRELIMINARIES

### 2.1 DIFFUSION-BASED OFFLINE REINFORCEMENT LEARNING

We consider a Markov Decision Process (MDP) defined by the tuple $(\mathcal{S}, \mathcal{A}, \mathcal{T}, r, \gamma)$, where $\mathcal{S}$ is the state space, $\mathcal{A}$ is the action space, $\mathcal{T}(\boldsymbol{s}'|\boldsymbol{s}, \boldsymbol{a})$ is the transition dynamics, $r(\boldsymbol{s}, \boldsymbol{a})$ is the reward function, and $\gamma \in (0, 1)$ is the discount factor. In offline RL, the agent is given a static dataset $\mathcal{D} \triangleq \{(\boldsymbol{s}, \boldsymbol{a}, r, \boldsymbol{s}')\}$, collected under an unknown behavior policy $\pi_b$, without environment access.

Diffusion-based offline RL (Mao et al., 2024; Wang et al., 2022; Fang et al., 2024; Chi et al., 2023; Kang et al., 2023) aims to learn a policy $\pi_\theta(a|s)$ parameterized as a conditional diffusion model that generates actions through a denoising process, *i.e.*,

$$\pi_\theta(\boldsymbol{a}|\boldsymbol{s}) = p_\theta(\boldsymbol{a}^{0:N}|\boldsymbol{s}) = \mathcal{N}(\boldsymbol{a}^N; \boldsymbol{0}, \boldsymbol{I}) \prod_{i=1}^{N} p_\theta(\boldsymbol{a}^{i-1}|\boldsymbol{a}^i, \boldsymbol{s}),$$

where $\boldsymbol{a}^i$ is an intermediate noisy action in the denoising chain. The denoising model $p_\theta(\boldsymbol{a}^{i-1}|\boldsymbol{a}^i, \boldsymbol{s})$ is modeled as a Gaussian distribution whose mean is predicted by a neural network. Following Ho et al. (2020), this network is typically implemented in the noise-prediction form as $\boldsymbol{\epsilon}_\theta$. We denote diffusion timesteps by superscripts $i \in \{1, \ldots, N\}$ and environment timesteps by subscripts $t \in \{1, \ldots, T\}$ to distinguish between the two temporal axes.

The actor is trained to minimize a hybrid loss that combines behavior cloning with critic guidance (Wang et al., 2022; Kang et al., 2023), *i.e.*,

$$\mathcal{L}_{\text{actor}}(\theta; \mathcal{D}) = \mathbb{E}_{(\boldsymbol{s}, \boldsymbol{a}) \sim \mathcal{D}, \boldsymbol{\epsilon} \sim \mathcal{N}(\boldsymbol{0}, \boldsymbol{I}), i \sim \text{Unif}[1, N]} \left[ \left\| \boldsymbol{\epsilon} - \boldsymbol{\epsilon}_\theta(\boldsymbol{a}^i, \boldsymbol{s}, i) \right\|^2 \right] - \alpha \cdot \mathbb{E}_{\boldsymbol{s} \sim \mathcal{D}, \boldsymbol{a}^0 \sim \pi_\theta}[Q_\phi(\boldsymbol{s}, \boldsymbol{a}^0)],$$

where $\boldsymbol{a}^0$ is the denoised action, and $\alpha$ balances imitation and value guidance Fang et al. (2024). Gradients from the critic are backpropagated through the entire denoising chain, steering the policy toward high-return actions. The critic $Q_\phi(\boldsymbol{s}, \boldsymbol{a})$ is trained via temporal-difference regression:

$$\mathcal{L}_{\text{TD}}(\phi; \mathcal{D}) = \mathbb{E}_{(\boldsymbol{s}, \boldsymbol{a}, r, \boldsymbol{s}') \sim \mathcal{D}} \left[ \left( r + \gamma \mathbb{E}_{\boldsymbol{a}' \sim \pi_\theta(\cdot|\boldsymbol{s}')}[Q_{\hat{\phi}}(\boldsymbol{s}', \boldsymbol{a}')] - Q_\phi(\boldsymbol{s}, \boldsymbol{a}) \right)^2 \right],$$

where $Q_{\hat{\phi}}(\boldsymbol{s}, \boldsymbol{a})$ is an ensemble-based or EMA-smoothed target network (Kang et al., 2023).

Together, the actor and critic form a diffusion-based offline RL agent in which the policy is both imitation-grounded and value-guided. This coupling makes diffusion policies expressive but also complicates unlearning, as both denoising dynamics and critic values must be modified consistently.

### 2.2 MACHINE UNLEARNING IN OFFLINE REINFORCEMENT LEARNING

Machine unlearning in offline RL aims to remove the influence of specific training data from a learned model without retraining from scratch (Bourtoule et al., 2021; Nguyen et al., 2022; Gong et al., 2024). Given offline data $\mathcal{D}$ and a training algorithm $\mathcal{R}$ that outputs a policy $\pi = \mathcal{R}(\mathcal{D})$, let $\mathcal{D}_f \subset \mathcal{D}$ be the data to be forgotten and $\mathcal{D}_r = \mathcal{D} \setminus \mathcal{D}_f$ the retained data. An unlearning algorithm $\mathcal{U}$ aims to produce an updated model $\pi' = \mathcal{U}(\pi, \mathcal{D}, \mathcal{D}_f)$ that produces predictions as if $\mathcal{D}_f$ had never contributed to training, an objective we interpret as removing the model's dependence on $\mathcal{D}_f$ while preserving performance on $\mathcal{D}_r$ and requiring significantly less computation than retraining from scratch.

In offline RL, the "as-if-unseen" criterion is inherently ill-defined: due to correlated trajectories, $\mathcal{D}_r$ often contains behaviors that overlap with or generalize those in $\mathcal{D}_f$ (see Fig. 3). As a result, a model trained on the full dataset may behave similarly to one retrained on $\mathcal{D}_r$ despite being influenced by $\mathcal{D}_f$. Therefore, in this work we adopt an influence-based view of unlearning: the unlearned model should not rely on $\mathcal{D}_f$, though its behaviors may reflect patterns present in the correlated retain set $\mathcal{D}_r$.

We distinguish two manifestations of unlearning. *Trajectory-level unlearning* focuses on removing the parameter-level influences of trajectories in $\mathcal{D}_f$, ensuring that their contribution cannot be identified

Figure 2: Overview of the RFF framework. RFF combines *critic unlearning*, which suppresses $Q$-values on unlearn data to eliminate reward incentives, and *actor unlearning*, which counteracts unlearn data via Fisher-scaled gradient updates. The denoising paths (left) show behavior shifting away from unlearn data while preserving high-value regions for retain data. The $Q$-landscape (right) highlights suppression of unlearned regions while maintaining support of desired behaviors.

even when correlated samples in $\mathcal{D}_r$ would otherwise reinforce similar behavior. *Behavior-level unlearning* targets the functional suppression of undesired modes (*e.g.*, "move backward" in Cheetah-dir) when these behaviors are uniquely supported by $\mathcal{D}_f$.

### 2.3 CHALLENGES IN UNLEARNING DIFFUSION-BASED OFFLINE REINFORCEMENT LEARNING

In diffusion-based offline RL, the dataset $\mathcal{D}$ is implicitly encoded into both the actor, through noise-conditioned regression aligned with the behavior policy and $Q$-guidance; and the critic, via Bellman updates. Policies are learned through a multi-step denoising process, where the influence of a single data point is distributed across the full diffusion process. Consequently, unlearning requires tracing and removing entangled influence across both the denoising dynamics and the critic's value estimation, making targeted removal difficult. Additional challenges arise from the supervision structure: reward signals shape the actor through $Q$-guidance, making credit assignment indirect and dependent on the full generative path. Moreover, because the actor and critic are jointly trained on sequential data, unlearning must preserve both their mutual consistency and long-horizon decision-making integrity to avoid utility degradation. These structural properties make unlearning diffusion policies more complex than in supervised or generative models. Effective solutions must therefore be noise-aware, reward-aligned, and coordinated across both actor and critic.

## 3 OFFLINE RL UNLEARNING VIA RELATIVE FISHER FORGETTING

We propose Relative Fisher Forgetting (RFF), the first principled framework for unlearning diffusion-based offline RL. RFF is designed to remove the influence of unlearning data while preserving useful knowledge. It consists of three complementary components: First, *actor unlearning* uses noise-aware forgetting gradients with relative Fisher preconditioning to forget specific samples from the diffusion policy. Second, *critic unlearning* suppresses the residual $Q$-values on forgotten data, preventing them from reinforcing undesirable behaviors. Third, *stabilization techniques* bound gradient magnitudes, alternate actor and critic updates, and incorporate retain-set regularization to preserve model utility. Together, they enable RFF to achieve reliable unlearning while maintaining retained data performance.

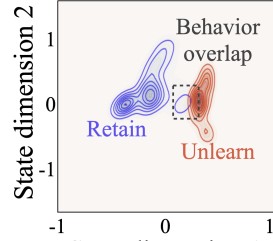

Figure 3: Unlearn and retain data overlap in state visitation of Ant-dir.

### 3.1 ACTOR UNLEARNING VIA RELATIVE FISHER WEIGHTING

The core idea of actor unlearning is to trace and cancel the influence of unlearning samples on the diffusion policy via a forgetting gradient. However, since diffusion policies spread the effect of each sample across the trajectory and unlearning and retaining sets may include overlapping or close samples (see Fig. 3), naïve updates often harm performance on retained data due to entangled influence. To mitigate this, we adopt a relative Fisher weighting strategy inspired by Foster et al.

(2024): parameters specialized to the unlearning data are updated more aggressively, while those critical to the retaining data are protected. This relative weighting prevents catastrophic forgetting of shared behaviors and enables targeted removal of forgotten skills.

**Unlearning Gradient.** Given a state-action pair $(\boldsymbol{s}, \boldsymbol{a})$ from the unlearning data $\mathcal{D}_{\mathrm{f}}$, we first construct a noisy action $\boldsymbol{a}^i$ by applying the forward diffusion at a randomly sampled step $i \in \{1, \ldots, N\}$. This aligns the unlearning update with the diffusion sampling process: in diffusion policies, training influence is distributed across all intermediate time steps. From this noisy input $\boldsymbol{a}^i$, the actor predicts the noise $\boldsymbol{\epsilon}_\theta(\boldsymbol{a}^i, \boldsymbol{s}, i)$ and recovers a multi-step denoised estimate $\hat{\boldsymbol{a}}^0$ by reversing the diffusion from timestep $i$. Using $\hat{\boldsymbol{a}}^0$ instead of the fully denoised action $\boldsymbol{a}^0$ ensures that unlearning is applied consistently throughout the generation path and prevents residual influence at earlier diffusion levels from regrowing forgotten behaviors. The per-sample loss is given by, $\ell_\theta(\boldsymbol{s}, \boldsymbol{a}, i, \boldsymbol{\epsilon}) = \left\| \boldsymbol{\epsilon} - \boldsymbol{\epsilon}_\theta(\boldsymbol{a}^i, \boldsymbol{s}, i) \right\|^2 - \alpha \cdot Q_\phi(\boldsymbol{s}, \hat{\boldsymbol{a}}^0)$, where the first term enforces denoising consistency and the second provides critic supervision on the predicted clean action. To compute the forgetting update, we take the expected gradient, *i.e.*,

$$\boldsymbol{g}_{\mathrm{f}}(\boldsymbol{s}, \boldsymbol{a}) := \mathbb{E}_{i \in \mathtt{Unif}[1,N], \boldsymbol{\epsilon} \sim \mathcal{N}(\boldsymbol{0}, \boldsymbol{I})} \left[ \nabla_\theta \ell_\theta(\boldsymbol{s}, \boldsymbol{a}, i, \boldsymbol{\epsilon}) \right], \tag{1}$$

which captures both the supervised denoising signal and the path-wise influence of the value term through the diffusion trajectory. In practice, we approximate this expectation using Monte Carlo samples over diffusion steps and noise, with truncated sampling for computational efficiency.

**Theoretical Motivation.** While the gradients in Eq. (1) provide directions for canceling the influence of $\mathcal{D}_{\mathrm{f}}$, directly applying them risks destabilizing parameters that are also important for retaining data. To prevent this, we are motivated to design a principled way to distinguish which parameters should be preserved during unlearning. Following prior work (Aich, 2021; Foster et al., 2024), we use Fisher information to quantify parameter importance on the retain data $\mathcal{D}_{\mathrm{r}}$. Parameters with high Fisher values strongly influence the loss on $\mathcal{D}_{\mathrm{r}}$ and should therefore undergo lower changes, whereas parameters with low Fisher information values can be safely adjusted to suppress the influence of $\mathcal{D}_{\mathrm{f}}$.

Let $\boldsymbol{g}_{\mathrm{f}} \triangleq \mathbb{E}_{(\boldsymbol{s}, \boldsymbol{a}) \in \mathcal{D}_{\mathrm{f}}}[\boldsymbol{g}_{\mathrm{f}}(\boldsymbol{s}, \boldsymbol{a})]$ denote the aggregated unlearning gradients over $\mathcal{D}_{\mathrm{f}}$. We formalize the unlearning step as searching for an update direction $\Delta_\theta$ that maximizes progress toward canceling the influence of $\mathcal{D}_{\mathrm{f}}$ while staying close to the Fisher-defined geometry of $\mathcal{D}_{\mathrm{r}}$, *i.e.*,

$$\max_{\Delta_\theta} \boldsymbol{g}_{\mathrm{f}}^\top \Delta_\theta, \quad \text{s.t. } \Delta_\theta^\top [F_\theta]^{\mathrm{r}} \Delta_\theta \le \rho,$$

where $[F_\theta]^{\mathrm{r}}$ is the empirical Fisher information matrix estimated on the retain data $\mathcal{D}_{\mathrm{r}}$. This constraint yields the closed-form solution, *i.e.*,

$$\Delta_\theta^* \propto [F_\theta]^{\mathrm{r}, -1} \boldsymbol{g}_{\mathrm{f}}, \tag{2}$$

which scales unlearning updates according to the inverse retain-data Fisher information, ensuring that unlearning does not harm behaviors supported by $\mathcal{D}_{\mathrm{r}}$.

**Relative Fisher Weighting.** Building on the optimization above, we approximate the Fisher geometry using the estimated diagonal Fisher information matrix (Aich, 2021; Foster et al., 2024), and compute Fisher diagonals for both the unlearning and retaining data,

$$[F_\theta]_j^{\mathrm{data}} = \mathbb{E}_{(\boldsymbol{s}, \boldsymbol{a}) \in \mathcal{D}_{\mathrm{data}}, i \in \mathtt{Unif}[1,N], \boldsymbol{\epsilon} \sim \mathcal{N}(\boldsymbol{0}, \boldsymbol{I})} \left[ \left( \frac{\partial \ell_\theta(\boldsymbol{s}, \boldsymbol{a}, i, \boldsymbol{\epsilon})}{\partial \theta_j} \right)^2 \right], \text{ where } \mathrm{data} \in \{\mathrm{f}, \mathrm{r}\}.$$

Here, $j$ indexes the neural network parameter $\boldsymbol{\epsilon}_\theta$. To ensure scalability, we approximate the Fisher diagonals using stochastic mini-batches, subsampled diffusion steps, and an exponential moving average of squared gradients. Using only the diagonal avoids $\mathcal{O}(d^2)$ storage and is sufficient because RFF relies on the relative contrast between unlearning and retaining Fisher scores (See Appx. 7.3).

In addition to the retain data Fisher in Eq. (2), we also incorporate the Fisher information computed on the unlearning data to identify parameters most influenced by $\mathcal{D}_{\mathrm{f}}$, enabling stronger forgetting in directions that the unlearning data has the greatest impact, *i.e.*,

$$\theta_j \leftarrow \theta_j + \eta \cdot \frac{[F_\theta]_j^{\mathrm{f}}}{[F_\theta]_j^{\mathrm{r}} + \epsilon'} [\boldsymbol{g}_{\mathrm{f}}]_j, \tag{3}$$

where $\eta$ controls the forgetting rate and $\epsilon' > 0$ ensures numerical stability. This update selectively amplifies forgetting on parameters associated with unlearning data while preserving those vital to retained behaviors, providing targeted suppression without utility collapse.

**Why Relative Fisher Weighting.** The relative Fisher ratio is necessary to isolate the influence of $\mathcal{D}_f$ from the shared structure present in $\mathcal{D}_r$. Using only the retain-set Fisher protects parameters important to $\mathcal{D}_r$ but offers little guidance on how the forgetting updates should be targeted on $\mathcal{D}_f$, while using only the forget-set Fisher over-updates parameters that also support retained behaviors. The ratio in Eq. (3) balances these effects: it amplifies forgetting in dimensions where $\mathcal{D}_f$ exerts disproportionate influence and attenuates updates where $\mathcal{D}_r$ provides important support. Empirically, variants using only forget-Fisher fail to achieve stable forgetting, whereas the relative form yields reliable suppression without degrading utility (See Appx. 7.4).

## 3.2 CRITIC UNLEARNING VIA VALUE SUPPRESSION

In addition to modifying the actor, it is essential to unlearn the influence of the forgetting data $\mathcal{D}_f$ on the critic $Q_\phi$. The critic not only provides guidance to the actor but also serves as the basis for offline evaluation. If unlearning samples remain associated with high $Q$-values, they can incentivize the policy to relearn forgotten behaviors and compromise unlearning fidelity. To address this, we introduce a value suppression mechanism that explicitly lowers the critic's estimates on unlearning data while maintaining reliable predictions on retained data.

The critic is trained with two complementary terms: *i)* TD regression on the retained data $\mathcal{D}_r$, *i.e.*,

$$\mathcal{L}_{\text{TD}}(\phi; \mathcal{D}_r) = \mathbb{E}_{(\boldsymbol{s}, \boldsymbol{a}, r, \boldsymbol{s}') \sim \mathcal{D}_r} \left[ \left( r + \gamma \mathbb{E}_{\boldsymbol{a}' \sim \pi_\theta(\cdot | \boldsymbol{s}')} [Q_{\hat{\phi}}(\boldsymbol{s}', \boldsymbol{a}')] - Q_\phi(\boldsymbol{s}, \boldsymbol{a}) \right)^2 \right].$$

Importantly, bootstrapping is performed only on retained samples, preventing unlearning data from leaking into the value targets. *ii)* Penalize high values for $\mathcal{D}_f$ using a hinge-style penalty, *i.e.*,

$$\mathcal{L}_{\text{sup}}(\phi; \mathcal{D}_f) = \mathbb{E}_{(\boldsymbol{s}, \boldsymbol{a}) \sim \mathcal{D}_f} [\max(0, Q_\phi(\boldsymbol{s}, \boldsymbol{a}) - \tau)].$$

Here, $\tau$ is a suppression floor calibrated from the retain data. Specifically, $\tau$ is chosen as a $q$-quantile of the TD targets on $\mathcal{D}_r$ with gradients stopped. This prevents the critic from artificially inflating retain estimates to minimize the penalty and ensures that unlearning samples are suppressed only relative to the retain data value scale. As a result, $Q$-values on $\mathcal{D}_f$ are pushed below a constant, retain-calibrated threshold rather than collapsed toward zero, preserving a bounded and smooth critic landscape (See empirical support in Appx. 7.4). The overall critic loss is,

$$\mathcal{L}_{\text{CU}}(\phi; \mathcal{D}_f, \mathcal{D}_r) \triangleq \lambda \mathcal{L}_{\text{TD}}(\phi; \mathcal{D}_r) + \mathcal{L}_{\text{sup}}(\phi; \mathcal{D}_f), \tag{4}$$

where $\lambda > 0$ controls the trade-off between stability and suppression strength. In practice, we warm up $\lambda$ and clip the gradient norm of the suppression term to avoid destabilizing updates. This combination ensures that the critic continues to learn accurate long-horizon returns on retained data while actively suppressing incentives for forgotten behaviors.

## 3.3 STABILIZING THE UNLEARNING PROCESS

RFF[2] performs unlearning by alternating actor and critic updates while incorporating several stabilization tricks. In each iteration, the actor parameters are updated using the relative Fisher weighted gradients in Eq. (3). The critic is then updated with the suppression-aware loss in Eq. (4), ensuring that forgotten samples are consistently devalued while retained dynamics are preserved. By alternating these, RFF decouples actor and critic dynamics and prevents destructive feedback loops.

To further stabilize training, we introduce three techniques. *Gradient norm control* clips the $L_2$ norm of aggregated unlearning gradients to prevent unstable parameter jumps. *Retain-set regularization* penalizes deviations from the pre-unlearning policy on $\mathcal{D}_r$, anchoring the policy to retained behaviors and preventing erosion of useful knowledge. *Convergence monitoring* tracks the divergence between the current and pre-unlearning policy on $\mathcal{D}_r$ as well as the suppression magnitude on $\mathcal{D}_f$; if either fails to stabilize, the forgetting rate or suppression coefficient is adaptively reduced. Together, these elements define RFF as an alternating actor–critic unlearning algorithm that selectively removes the influence of unlearning samples while preserving stability and performance on retained data.

---

[2]See Appx. 7.2 for complete algorithm.

| Tasks | Data | Retrain | Fine-tuning | Random-reward | TrajDeleter | EraseDiff | RFF (Ours) |
|---|---|---|---|---|---|---|---|
| Hopper | $\mathcal{D}_r$ | 85.0% | 84.2% | 86.1% | 85.4% | 83.0% | 85.1% |
| | $\mathcal{D}_{f,1\%}$ | 85.1% | 27.6% | 28.7% | 12.1% | 14.9% | **10.1%** |
| | $\mathcal{D}_{f,5\%}$ | 84.8% | 24.7% | 28.8% | 13.0% | 14.7% | **11.0%** |
| HalfCheetah | $\mathcal{D}_r$ | 93.4% | 93.3% | 92.3% | 93.4% | 93.3% | 92.6% |
| | $\mathcal{D}_{f,1\%}$ | 92.4% | 58.3% | 69.4% | 6.2% | 6.0% | **4.3%** |
| | $\mathcal{D}_{f,5\%}$ | 93.1% | 32.3% | 56.4% | 5.7% | 5.4% | **3.3%** |
| Walker2D | $\mathcal{D}_r$ | 95.5% | 94.7% | 94.8% | 93.6% | 96.2% | 94.8% |
| | $\mathcal{D}_{f,1\%}$ | 95.3% | 57.3% | 61.7% | 12.6% | 10.8% | **8.8%** |
| | $\mathcal{D}_{f,5\%}$ | 95.5% | 45.3% | 58.4% | 9.4% | 13.7% | **8.7%** |

Table 1: Percentages of positive predictions by TrajAuditor for the unlearn data post unlearning. $\mathcal{D}_r$ indicates that the retaining dataset is not subjected to unlearning. The $\mathcal{D}_{f,1\%}$ and $\mathcal{D}_{f,5\%}$ denote the unlearning dataset with the size of 1% and 5% of the original dataset. On $\mathcal{D}_r$ the higher positive predictions the better, and on $\mathcal{D}_f$, the lower the better.

## 4 EXPERIMENT

Our experiments address three key questions: *(Q1)* Does RFF effectively remove the influence of designated trajectories or behaviors while preserving overall task performance? *(Q2)* How does RFF compare to prior unlearning baselines in utility retention, and what is the efficiency cost? *(Q3)* What is the contribution of actor *vs.* critic unlearning, and how does RFF scale with unlearning ratio?

### 4.1 EXPERIMENT SETUP

**Tasks and Datasets.** We evaluate on both single-task and multi-task offline RL benchmarks to test trajectory-level and behavior-level unlearning. For single-task settings, we use MuJoCo control environments (Todorov et al., 2012) (Hopper, HalfCheetah, Walker2D) with D4RL medium-expert datasets (Fu et al., 2020), following Gong et al. (2024). For each task, we randomly sample 1% and 5% of the training data to unlearn. For multi-task settings, we adopt Cheetah-Dir, Cheetah-Vel, and Ant-Dir, which require mastering multiple behaviors and naturally test behavior-level unlearning (*e.g.*, forgetting "moving east"). Following prior work (Xu et al., 2022), datasets are collected using SAC-trained policies. In each environment, we randomly sample one task as the behavior to unlearn, corresponding to 50%, 2.5% and 2% of the data in Cheetah-Dir, Cheetah-Vel, and Ant-Dir, respectively. Additional implementation details are provided in Appx. 7.3.

**Baselines.** In this work, we focus on unlearning diffusion policies (Wang et al., 2022), as they are effective and widely adopted in offline RL. Extending our framework to other diffusion-based approaches (Chi et al., 2023; Mao et al., 2024; Janner et al., 2022) is left for future work.

We compare RFF against four baselines following Gong et al. (2024): *Retrain:* training the agent from scratch on $\mathcal{D}_r$. It is resource-intensive but works as a reference point. *Fine-tuning:* continue training on $\mathcal{D}_r$ with a limited number of iterations. *Random-reward:* replace rewards in $\mathcal{D}_f$ with random values sampled from the dataset's reward range, followed by fine-tuning on the modified data. *TrajDeleter (Gong et al., 2024):* minimizes the $Q$-values on $\mathcal{D}_f$ while maximizing those on $\mathcal{D}_r$. Additionally, we include one state-of-the-art diffusion unlearning approach: *EraseDiff (Wu et al., 2024):* perturb score estimates on $\mathcal{D}_f$ while regularizing on $\mathcal{D}_r$. All methods start from the same pre-trained diffusion policy and critic. For fairness, we match total update budgets across methods (see Appx. 7.3).

**Evaluation Metrics.** We evaluate along three dimensions: *Forgetting Effectiveness:* measured by the TrajAuditor score (Gong et al., 2024), where lower positive prediction rates indicate stronger forgetting, and by the return drop on forgotten behaviors. *Utility retention:* normalized returns after unlearning for single-tasks and average normalized returns across retained tasks in multi-tasks. *Efficiency*: wall-clock runtime and the number of gradient steps relative to retraining.

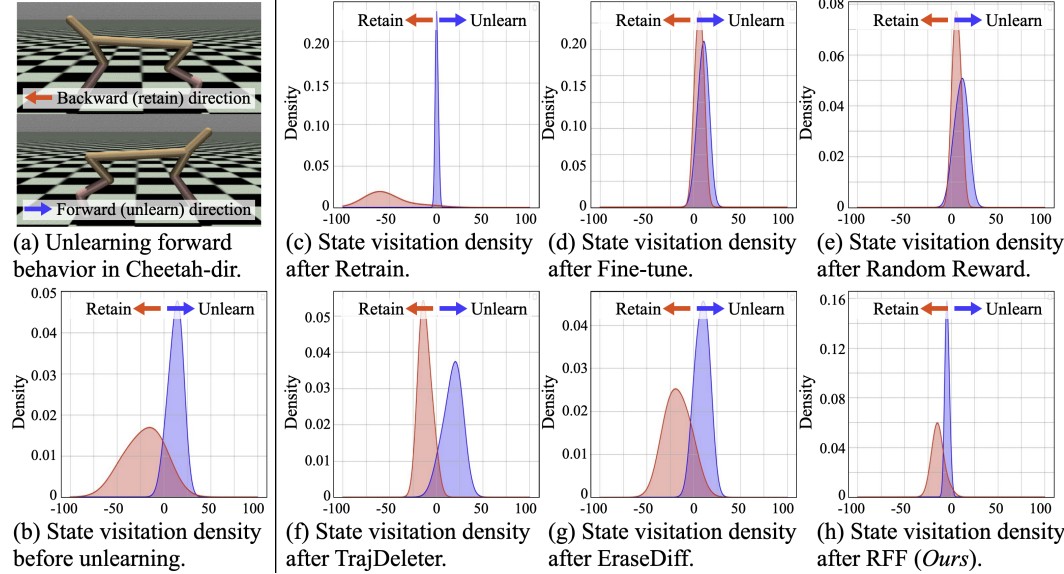

Figure 4: State visitation densities in Cheetah-Dir. Our RFF effectively forgets the unlearn forward direction while preserving the retain backward direction.

### 4.2 UNLEARNING PERFORMANCE ANALYSIS

**Trajectory-level Unlearning.** Tab. 1 reports the unlearning effectiveness on Hopper, HalfCheetah, and Walker2D. Retrain does not completely erase $\mathcal{D}_f$, as strong correlations between the retain and forget trajectories causing the model to revisit undesired states, consistent with prior observations (Jiang et al., 2024). This does not contradict unlearning; rather, it highlights that the classical "as-if-unseen" criterion is ill-posed in correlated offline RL settings. Fine-tuning and Random Reward reduce predictions somewhat but often leave high auditor scores on the forget data, showing that shallow updates cannot reliably remove unlearn data. TrajDeleter reduces predictions more substantially but exhibits inconsistent forgetting across tasks and works better when unlearning HalfCheetah. EraseDiff improves forgetting in HalfCheetah but still suffers from instability. This is likely because EraseDiff cannot fully remove the influence of the $Q$-values that assign high values on unlearn data. In contrast, our RFF achieves the lowest prediction rates on unlearning data while preserving consistent behavior with retaining data, demonstrating both effective and robust unlearning.

**Behavior-level Unlearning.** Fig. 1 illustrates Ant-Dir the state visitation before and after unlearning. Before unlearning, the agent explores all directions, producing nearly uniform $360°$ coverage. After applying RFF, behaviors associated with the unlearn direction are effectively suppressed while retain directions remain intact. Baselines fail to achieve a comparable disentanglement: Fine-tuning and Random Reward often leave traces of the unlearn behavior, and even Retrain exhibits partial drift toward the undesired direction due to the correlations between tasks. In addition, Fig. 6(c) also validates the superior unlearning performance of RFF: it leads to the most return decrease along the undesirable behavior while maintaining satisfactory return on retain behaviors. Consistent observations are made in the Cheetah-Dir and Cheetah-Vel (see Fig. 4 and Fig. 7 in Appx. 7.4).

### 4.3 UTILITY & EFFICIENCY COMPARISON

**Utility Evaluation.** *Trajectory-level Unlearning.* Fig. 5 reports normalized returns before and after unlearning. Across Hopper, HalfCheetah, and Walker2D, RFF consistently preserves higher returns on the retain data compared to baselines. This indicates that RFF is able to maintain unlearned model utility. *Behavior-level Unlearning.* Fig. 6 shows return changes on retain and unlearn behaviors. Among all baselines, RFF best preserves performance on retained behaviors (small return change) while most strongly suppressing the unlearn behaviors (large return drop). These results highlight RFF's ability to selectively remove targeted behavior modes while avoiding collateral degradation. In Cheetah-Dir and Ant-Dir, simpler baselines including Retrain, Fine-tuning, and Random Reward suppress the unlearn mode but also reduce performance on retain behavior. Their degrada-

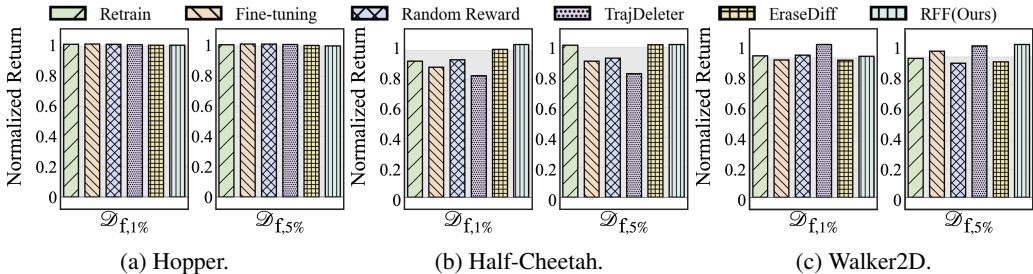

Figure 5: Normalized returns on Hopper, Half-Cheetah, Walker2D after unlearning. Results are shown for forgetting ratios $\mathcal{D}_{f,1\%}$ and $\mathcal{D}_{f,5\%}$, highlighting that RFF maintains high performance on retain data compared to baselines.

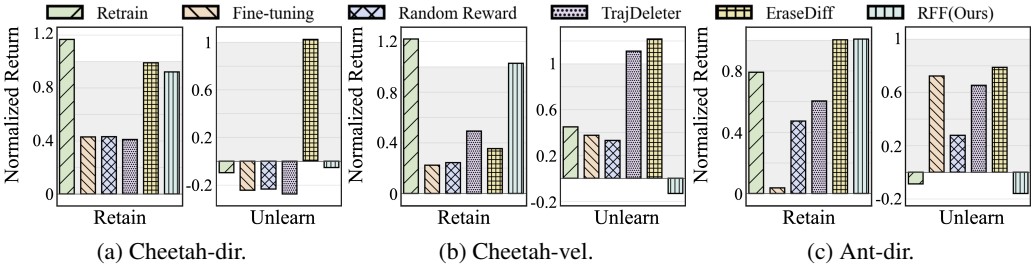

Figure 6: Normalized return on Cheetah-Dir, Cheetah-Vel, and Ant-Dir. Higher returns on retain tasks indicate better utility, while lower returns on unlearn tasks reflect stronger forgetting. Grey background bars show the average normalized returns of retain/unlearn tasks.

tion is particularly severe in Cheetah-vel, where behaviors correspond to different forward speeds: forgetting one speed (*e.g.*, $2m/s$) conflicts with retaining others, leading to collapse across all tasks.

TrajDeleter and EraseDiff also struggle because they address forgetting in only one component of the actor–critic architecture, leaving residual influence in either the policy or the value function. In contrast, RFF jointly unlearns both, yielding more faithful forgetting with minimal loss of utility.

**Efficiency Evaluation.** Tab. 2 reports runtime and gradient steps for Ant-Dir. Retraining is prohibitively expensive, requiring over 100k gradient steps. All approximate unlearning methods are substantially more efficient, with RFF requiring only 75 steps while achieving superior forgetting and utility preservation. Although alternating actor–critic updates introduce slight overhead, RFF remains almost $107\times$ faster than retraining and more effective than competing baselines. More results in Appx. 7.4.

| Method | Time | GS |
|---|---|---|
| Retrain | 364 min | 100k |
| Finetuning | 26 min | 200 |
| Random Reward | 26 min | 200 |
| TrajDeleter | 30 min | 75 |
| EraseDiff | 26 min | 75 |
| RFF (Ours) | 29 min | 75 |

Table 2: Runtime & gradient steps (GS) in Ant-Dir.

### 4.4 ABLATION ON ACTOR, CRITIC AND SCALABILITY

**Ablation on Actor & Critic Unlearning.** To analyze forgetting efficacy, we ablate RFF into three variants: RFF(A✓/C✗) which disables critic unlearning, RFF(A✗/C✓) which disables actor unlearning, and RFF- which removes the stabilization mechanism. For each variant, we measure two metrics on the Ant-dir task: the retain performance change $\Delta R_r$, defined as the return difference before and after unlearning (lower is better, indicating less degradation on retain data), and the unlearning success $\Delta R_f$, defined as the return difference before and after unlearning on the forget set (higher is better, indicating more effective forgetting).

Tab. 3 shows that both actor and critic unlearning, together with stabilization, are crucial for achieving effective forgetting while preserving performance. When critic unlearning is removed (RFF(A✓/C✗)), the retain performance $\Delta R_r$ remains stable (0.5, lowest among variants), but unlearning success $\Delta R_f$ is poor (52.5), indicating that the critic continues to reinforce undesired behaviors. Conversely, without actor unlearning (RFF(A✗/C✓)), the critic suppresses rewards but the policy still imitates the unlearn trajectories, yielding higher unlearning success but large degradation on retain data. Removing stabilization (RFF-) causes the critic to undervalue retain data, resulting in severe performance loss

| Variants | $\Delta R_r\downarrow$ | $\Delta R_f\uparrow$ |
|---|---|---|
| RFF(A✓/C✗) | **0.5** | 52.5 |
| RFF(A✗/C✓) | 69.3 | 86.1 |
| RFF- | 58.7 | 508.3 |
| RFF(A✓/C✓) | 44.9 | **514.2** |

Table 3: Ablation results in Ant-Dir.

| Unlearning Ratio | Metrics | Retrain | TrajDeleter | EraseDiff | RFF (Ours) |
|---|---|---|---|---|---|
| 25% | $\hat{R}$ | 0.84 | 0.84 | **0.87** | 0.86 |
| | $\mathcal{D}_{f,25\%}$ | 48.4% | 7.1% | 11.4% | **6.7%** |
| 50% | $\hat{R}$ | 0.80 | 0.84 | **0.86** | **0.86** |
| | $\mathcal{D}_{f,50\%}$ | 5.2% | 3.9% | 12.5% | **3.8%** |

Table 4: Utility (normalized return $\hat{R}$, ↑ better) *vs.* unlearning efficacy (TrajAuditor positives, ↓ better) at 25%, 50% unlearning ratios.

despite good forgetting. The full method (RFF(A✓/C✓)) achieves the best balance, *i.e.*, strong forgetting and controlled degradation on retain data. These results confirm that actor unlearning, critic unlearning, and stabilization are complementary and jointly necessary for robust unlearning.

**Scalability.** To assess the scalability of RFF, we further evaluate RFF's performance under substantially larger unlearning ratios of 25% and 50% on Hopper (Tab. 4). Across both settings, RFF maintains strong forgetting and is able to achieve the lowest TrajAuditor scores while preserving competitive normalized returns. This demonstrates that its unlearning dynamics remain stable even when a significant fraction of the dataset is removed. In contrast, retraining exhibits high positive prediction rates (48.4% at 25% unlearning), reflecting the strong correlation structure of the dataset and the diffusion policy's inherent generalization across overlapping state–action regions. This behavior indicates that the classical "as-if-never-seen" criterion is inherently ill-posed for diffusion policies, given their tendency to generalize forgotten behaviors from correlated retain data. RFF directly addresses this challenge by explicitly suppressing both actor and critic influence from the forget set, enabling more faithful and robust unlearning than baselines under large unlearning ratios.

## 5 RELATED WORK

Machine unlearning has been studied in supervised learning via retraining (Bourtoule et al., 2021), influence functions (Guo et al., 2019), and distillation (Golatkar et al., 2020), and extended to generative models through gradient editing or distribution re-weighting (Gandikota et al., 2023; Kumari et al., 2023; Wu et al., 2024). These methods ignore the behavioral dynamics of RL. Safe RL mitigates undesirable actions through constraints or reward shaping (García & Fernández, 2015; Ray et al., 2019), but focuses on prevention rather than post-hoc removal. Early RL unlearning approaches revoke entire environments (Ye et al., 2023), and TrajDeleter (Gong et al., 2024) targets trajectory-level forgetting but leaves actor incentives intact. In contrast, we address selective trajectory- and behavior-level unlearning in diffusion-based offline RL by jointly updating actor and critic while preserving retain performance. (See Appx. 7.5 for more information.)

## 6 CONCLUSION AND LIMITATIONS

We presented Relative Fisher Forgetting (RFF), a principled framework for unlearning in diffusion-based offline RL. RFF leverages relative Fisher information to selectively suppress the influence of undesired trajectories while preserving retained knowledge. By jointly unlearning actor and critic components and incorporating stabilization, RFF achieves effective forgetting with minimal performance degradation. Experiments across trajectory- and behavior-level settings show that RFF outperforms retraining and prior unlearning baselines in efficiency, fidelity, and stability, providing the first systematic solution for unlearning in diffusion policies.

Despite these advances, several limitations remain. RFF assumes explicit access to the forget set, which may be difficult to identify in practice. Our empirical evaluation is conducted in simulated continuous-control environments; while these controlled settings allow us to isolate algorithmic behavior, further validation on real-world offline RL datasets is needed. Many real applications (*e.g.*, autonomous driving logs, robotic teleoperation demonstrations, and user-generated interaction traces) exhibit similar correlated trajectory structure and post-hoc correction requirements, suggesting that RFF's influence-based unlearning principles transfer naturally beyond simulation. Moreover, while RFF is empirically effective, developing provable guarantees under distribution shifts or partial data deletion requests remains an important direction of future work. We believe this work lays the foundation for trustworthy unlearning in decision-making systems, enabling safer, more adaptive, and more accountable RL agents in real-world deployment.

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
