# 7 APPENDIX

## 7.1 ETHICS AND REPRODUCIBILITY STATEMENTS

**Ethics Statement.** This research adheres to the ICLR Code of Ethics. All experiments are conducted on publicly available benchmark datasets and environments that do not involve human subjects or personally identifiable information. No new data collection was performed, and no sensitive or private information is included. The contributions of this work are methodological, aiming to advance machine learning. While reinforcement learning and related methods may be applied in safety-critical or socially sensitive domains, this paper does not directly address or deploy in such contexts. We have taken care to report results honestly, acknowledge limitations, and follow best practices for research integrity. No conflicts of interest or ethical concerns beyond standard research conduct arise from this work.

**Reproducibility Statement.** We have made every effort to ensure the reproducibility of our results. Detailed descriptions of the proposed algorithms, theoretical assumptions, and derivations are provided in the main text and appendices. Hyperparameter settings, model architectures, and training configurations are reported in full. Data preprocessing procedures and evaluation protocols are documented, and all datasets used are publicly available. An anonymous link to source code and instructions for reproducing experiments is included in the paper. Together, these resources ensure that independent researchers can reproduce and verify our findings.

**LLM Usage Statement.** We used OpenAI's GPT-5 solely to improve clarity and readability of the writing; all research ideas, methods, analyses, and results are the authors' own.

## 7.2 ALGORITHM

We introduce Relative Fisher Forgetting (RFF) in Alg. 1, an alternating actor–critic procedure designed to selectively remove the influence of unlearning data while retaining policy competence. RFF operates in three coordinated stages: (i) the actor-side update (Alg. 3) leverages a relative Fisher preconditioner that scales forgetting gradients on unlearning data against their retained counterparts, thereby suppressing harmful behaviors while anchoring the policy to its pre-unlearn state; (ii) the critic-side update (Alg. 4) enforces selective conservatism by penalizing $Q$-values on unlearned trajectories relative to retained ones, ensuring the value function no longer incentivizes forgotten behaviors; and (iii) a lightweight convergence adapter (Alg. 5) monitors policy drift and value suppression, automatically adjusting hyperparameters to stabilize training. This modular design allows RFF to achieve effective forgetting without full retraining, balancing utility preservation, stability, and computational efficiency.

---

**Algorithm 1** Relative Fisher Forgetting.

---

**Require:** Retain data $\mathcal{D}_r$, Unlearn data $\mathcal{D}_f$; actor $\pi_\theta$, critic $Q_\phi$, target critic $Q_{\hat{\phi}}$
**Require:** Hyperparams: $\eta, \lambda, \epsilon' > 0, (k_\pi, k_Q), \beta_F, (\delta_\pi, \delta_Q), \tau_{\text{target}}, \alpha$
  1: Initialize EMA states $M_f=\mathbf{0}, M_r=\mathbf{0}, t_F=0$             ▷ Fishers for actor
  2: **while** not converged **do**
  3:     ACTORRFF($\pi_\theta, \mathcal{D}_f, \mathcal{D}_r, \eta, \epsilon', \beta_F, M_f, M_r, t_F, k_\pi$)
  4:     CRITICUNLEARNING($Q_\phi, Q_{\hat{\phi}}, \pi_\theta, \mathcal{D}_r, \mathcal{D}_f, \lambda, \delta_Q, \tau_{\text{target}}, k_Q$)
  5:     CONVERGENCEADAPTER($\pi_\theta, \pi_{\theta^{\text{pre}}}, Q_\phi, \mathcal{D}_r, \mathcal{D}_f, \delta_\pi, \delta_Q, \alpha$)     ▷ Adjust knobs
  6: **end while**
  7: **return** $\pi_\theta, Q_\phi$

---

## 7.3 IMPLEMENTATION DETAILS

**Efficient Fisher Estimation.** While the diagonal Fisher information matrix offers a tractable approximation to local parameter curvature, computing it explicitly across all parameters in diffusion models remains computationally intensive. Therefore, we use online estimation during training

---

**Algorithm 2** UPDATEFISHERS (bias-corrected diagonal EMAs)

---

**Require:** Batches $\mathcal{B}_\mathrm{f} \subset \mathcal{D}_\mathrm{f}$, $\mathcal{B}_\mathrm{r} \subset \mathcal{D}_\mathrm{r}$, actor $\pi_\theta$, EMA states $M_\mathrm{f}, M_\mathrm{r}, t_F$, factor $\beta_F$
1: $\boldsymbol{g}_\mathrm{f} \leftarrow \nabla_\theta \frac{1}{|\mathcal{B}_\mathrm{f}|} \sum_{(\boldsymbol{s},\boldsymbol{a}) \in \mathcal{B}_\mathrm{f}} \ell_\theta(\boldsymbol{s}, \boldsymbol{a}, i, \boldsymbol{\epsilon})$
2: $\boldsymbol{g}_\mathrm{r} \leftarrow \nabla_\theta \frac{1}{|\mathcal{B}_\mathrm{r}|} \sum_{(\boldsymbol{s},\boldsymbol{a}) \in \mathcal{B}_\mathrm{r}} \ell_\theta(\boldsymbol{s}, \boldsymbol{a}, i, \boldsymbol{\epsilon})$
3: $M_\mathrm{f} \leftarrow \beta_F M_\mathrm{f} + (1-\beta_F) \boldsymbol{g}_\mathrm{f}^{\odot 2}; \quad M_\mathrm{r} \leftarrow \beta_F M_\mathrm{r} + (1-\beta_F) \boldsymbol{g}_\mathrm{r}^{\odot 2}; \quad t_F \mathrel{+}= 1$
4: $F^\mathrm{f} \leftarrow M_\mathrm{f}/(1-\beta_F^{t_F}); \quad F_\mathrm{r} \leftarrow M_\mathrm{r}/(1-\beta_F^{t_F})$
5: **return** $F^\mathrm{f}, F^\mathrm{r}, M_\mathrm{f}, M_\mathrm{r}, t_F$

---

**Algorithm 3** ACTORRFF (Actor Unlearning)

---

**Require:** $\pi_\theta$, datasets $\mathcal{D}_\mathrm{f}, \mathcal{D}_\mathrm{r}$; $\eta, \epsilon', \beta_F$; EMA states $M_\mathrm{f}, M_\mathrm{r}, t_F$; steps $k_\pi$
1: **for** $t = 1$ to $k_\pi$ **do**
2: $\quad$ Sample $\mathcal{B}_\mathrm{f} \subset \mathcal{D}_\mathrm{f}$, $\mathcal{B}_\mathrm{r} \subset \mathcal{D}_\mathrm{r}$
3: $\quad (F^\mathrm{f}, F^\mathrm{r}, M_\mathrm{f}, M_\mathrm{r}, t_F) \leftarrow$ UPDATEFISHERS$(\mathcal{B}_\mathrm{f}, \mathcal{B}_\mathrm{r}, \pi_\theta, M_\mathrm{f}, M_\mathrm{r}, t_F, \beta_F)$
4: $\quad \boldsymbol{g}_\mathrm{f} \leftarrow \nabla_\theta \frac{1}{|\mathcal{B}_\mathrm{f}|} \sum_{(\boldsymbol{s},\boldsymbol{a}) \in \mathcal{B}_\mathrm{f}} \ell_\theta(\boldsymbol{s}, \boldsymbol{a}, i, \boldsymbol{\epsilon});$
5: $\quad W \leftarrow F^\mathrm{f} \oslash (F^\mathrm{r} + \epsilon'); \quad \Delta\theta \leftarrow \eta(W \odot \boldsymbol{g}_\mathrm{f})$
6: $\quad \theta \leftarrow \theta + \Delta\theta$
7: **end for**
8: **return** $\pi_\theta, M_\mathrm{f}, M_\mathrm{r}, t_F$

---

and subsampled Fisher estimation to reduce the computational burden of estimating the Fisher matrix (Foster et al., 2024) in the actor unlearning, without sacrificing its regularization benefits.

Instead of computing the Fisher matrix post hoc, we adopt an online estimation strategy by tracking an exponential moving average (EMA) of squared gradients during the initial training phase. Specifically, for each model parameter $\theta_j$, we maintain: $[F_\theta]_j \leftarrow \beta \cdot [F_\theta]_j + (1 - \beta) \cdot \left(\frac{\partial \ell_\theta}{\partial \theta_j}\right)^2$, where $\beta \in [0, 1)$ is a smoothing coefficient. This approach amortizes the cost of Fisher estimation across training and requires no additional backpropagation passes.

To further reduce overhead, we estimate the Fisher matrix using a small, randomly selected subset of the unlearning dataset $\mathcal{D}_\mathrm{f}$. For each selected sample, we compute gradients at only a few diffusion steps $i$ and noise samples $\boldsymbol{\epsilon} \sim \mathcal{N}(\boldsymbol{0}, \boldsymbol{I})$, rather than integrating over the full diffusion horizon. This Monte Carlo approximation maintains a good fidelity of the diagonal Fisher estimate while drastically reducing computation.

Since actor unlearning uses Fisher only to capture the relative importance of parameters across unlearning and retaining sets, approximate estimation is sufficient: what matters is the contrast between the two sets, not the exact value of each Fisher term.

**Implementation Setup.** All results are averaged across the same random seeds and reported with normalized scores. Experiments were conducted on a single NVIDIA RTX 6000 Ada GPU (48GB) using PyTorch and Gymnasium libraries. We evaluate our unlearning process using the MuJoCo benchmark (Todorov et al., 2012), chosen for its complexity and wide adoption in prior unlearning studies. This benchmark is particularly well-suited for testing generalized forgetting in diffusion-based policies.

**MuJoCo Multi-Task Agents.** Following the multi-task protocol of Prompt-DT (Xu et al., 2022), we evaluate on three locomotion environments: **Cheetah-dir**, **Cheetah-vel**, and **Ant-dir**. These tasks penalize deviations from target goals and are designed to assess task-specific unlearning in few-shot settings.

- **Cheetah-dir:** Two tasks (forward, backward). Reward is proportional to velocity along the target direction.
- **Cheetah-vel:** 10 tasks, each with a target velocity. Reward penalizes the $\ell_2$ distance to the target.
- **Ant-dir:** 25 tasks, each with a target direction. Reward is proportional to velocity along the goal direction.

---

**Algorithm 4** CRITICUNLEARNING

**Require:** $Q_\phi, Q_{\hat\phi}, \pi_\theta$; datasets $\mathcal{D}_r, \mathcal{D}_f$; $\lambda, \delta_Q, \tau_{\text{target}}$; steps $k_Q$

1: **for** $t = 1$ to $k_Q$ **do**
2:      Sample $\mathcal{B}_r \subset \mathcal{D}_r$, $\mathcal{B}_f \subset \mathcal{D}_f$
3:      $y(\boldsymbol{s}, \boldsymbol{a}, r, \boldsymbol{s}') \leftarrow r + \gamma\, \mathbb{E}_{\boldsymbol{a}' \sim \pi_\theta(\cdot|\boldsymbol{s}')}\big[Q_{\hat\phi}(\boldsymbol{s}', \boldsymbol{a}')\big]$
4:      $\mathcal{L}_{\text{TD}} \leftarrow \frac{1}{|\mathcal{B}_r|}\sum_{(\boldsymbol{s},\boldsymbol{a},r,\boldsymbol{s}') \in \mathcal{B}_t}(Q_\phi(\boldsymbol{s},\boldsymbol{a}) - y)^2$
5:      $\bar{q}_r \leftarrow \text{stopgrad}\Big(\frac{1}{|\mathcal{B}_r|}\sum_{(\boldsymbol{s},\boldsymbol{a}) \in \mathcal{B}_r} Q_\phi(\boldsymbol{s},\boldsymbol{a})\Big)$
6:      $\tau \leftarrow \bar{q}_r - \delta_Q$
7:      $\mathcal{L}_{\text{sup}} \leftarrow \frac{1}{|\mathcal{B}_f|}\sum_{(\boldsymbol{s},\boldsymbol{a}) \in \mathcal{B}_f} \max\big(0, Q_\phi(\boldsymbol{s},a) - \tau\big)$
8:      $\phi \leftarrow \phi - \nabla_\phi\big(\lambda \cdot \mathcal{L}_{\text{TD}} + \mathcal{L}_{\text{sup}}\big)$
9:      $\hat\phi \leftarrow \tau_{\text{target}}\,\phi + (1 - \tau_{\text{target}})\,\hat\phi$
10: **end for**
11: **return** $Q_\phi, Q_{\hat\phi}$

---

**Algorithm 5** CONVERGENCEADAPTER

**Require:** $\pi_\theta, \pi_{\theta^{\text{pre}}}, Q_\phi$; datasets $\mathcal{D}_r, \mathcal{D}_f$ ; $\delta_\pi, \delta_Q, \alpha$

1: Calculate policy drift $D_\pi \leftarrow \mathbb{E}_{(\boldsymbol{s},\boldsymbol{a}) \in \mathcal{D}_r}\big[\|\pi_\theta(\boldsymbol{a}|\boldsymbol{s}) - \pi_{\theta^{\text{pre}}}(\boldsymbol{a}|\boldsymbol{s})\|\big]$
2: Calculate value gap $D_Q \leftarrow \mathbb{E}_{(\boldsymbol{s},\boldsymbol{a}) \in \mathcal{D}_f}[Q_\phi(\boldsymbol{s},\boldsymbol{a})] - \mathbb{E}_{(\boldsymbol{s},\boldsymbol{a}) \in \mathcal{D}_r}[Q_\phi(\boldsymbol{s},\boldsymbol{a})]$
3: **if** $D_\pi > \delta_\pi$ **then**
4:      Update $\eta \leftarrow \frac{\eta}{1 + \alpha D_\pi}$
5: **end if**
6: **if** $D_Q > -\delta_Q$ **then**
7:      Update $\lambda \leftarrow \frac{\lambda}{1 + \alpha D_Q}$
8: **end if**
9: **return** $\eta, \lambda$ for automatic tuning

---

**MuJoCo Single-Task Agents.** Following TrajDeleter (Gong et al., 2024), we evaluate on three locomotion environments: **Hopper**, **HalfCheetah**, and **Walker2d**. These tasks use built-in Gymnasium return functions, where higher directional velocity yields higher returns.

**Pretraining.** For multi-task agents, fine-tuning was necessary to avoid task imbalance:

- **Cheetah-dir:** Equal split of forward/backward led to forward bias. Fine-tuning on backward-only tasks for at least 1k batches (batch size 2048) corrected performance.
- **Cheetah-vel:** Equal split across tasks led to poor performance at high velocities. Fine-tuning on the fastest 5 velocities for 1k batches improved stability.
- **Ant-dir:** Pretraining on tasks provided sufficient performance without extra fine-tuning.

For single-task agents, full dataset pretraining was sufficient for stable unlearning without fine-tuning.

**Dataset Splits & Hyperparameters.** Dataset split details and hyperparameters are provided in Tab. 5 and Tab. 7 respectively.

| Environment | Training Set | Unlearning Set |
|---|---|---|
| Cheetah-dir | 2 tasks [0, 1] | 2 tasks [0, 1] |
| Cheetah-vel | 10 tasks [0, 4, 8, 12, 16, 20, 24, 28, 32, 36] | 1 task [28] |
| Ant-dir | 25 tasks [0:2:48] | 1 task [40] |
| Hopper | 999,494 samples ($\sim$4998 traj.) | 1% ($\sim$50 traj.) / 5% ($\sim$250 traj.) |
| HalfCheetah | 1M samples (5000 traj.) | 1% (50 traj.) / 5% (250 traj.) |
| Walker2d | 999,214 samples ($\sim$4997 traj.) | 1% ($\sim$50 traj.) / 5% ($\sim$250 traj.) |

Table 5: Training and unlearning dataset splits for multi-task and single-task MuJoCo benchmarks.

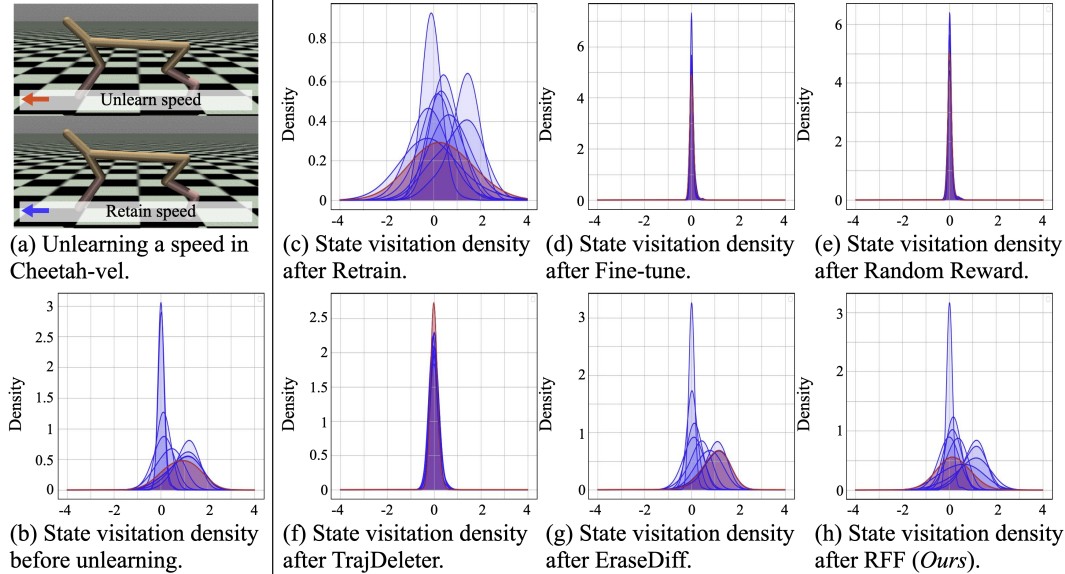

Figure 7: State visitation densities in Cheetah-vel. Our RFF effectively forgets the designated unlearn speed while preserving the retain speeds.

| Methods | Tasks | | | | | |
|---|---|---|---|---|---|---|
| | Hopper | | HalfCheetah | | Walker2D | |
| | $\mathcal{D}_{\text{f},1\%}$ | $\mathcal{D}_{\text{f},5\%}$ | $\mathcal{D}_{\text{f},1\%}$ | $\mathcal{D}_{\text{f},5\%}$ | $\mathcal{D}_{\text{f},1\%}$ | $\mathcal{D}_{\text{f},5\%}$ |
| Retraining | 267 min | 274 min | 269 min | 281 min | 272 min | 277 min |
| Fine-tuning | 0.52 min | 0.52 min | 0.52 min | 0.52 min | 0.50 min | 0.50 min |
| Random-reward | 0.52 min | 0.52 min | 0.52 min | 0.52 min | 0.52 min | 0.52 min |
| TrajDeleter | 9 min | 9 min | 8 min | 8 min | 9 min | 9 mine |
| EraseDiff | 6 min | 6 min | 6 min | 6 min | 6 min | 6 min |
| RFF | 7 min | 7 min | 10 min | 10 min | 11 min | 11 min |

Table 6: Training duration (minutes) across tasks for different unlearning baselines and our method. Retraining is reported at 0%, 1%, and 5% duration.

### 7.4 More Experiment Results

**Behavior-unlearning Performance.** Fig. 7 shows the unlearning result of Cheetah-vel. It shows that RFF is able to unlearn the specific unlearning speed while maintaining the retain speed well. Since this unlearning task involves correlated behaviors, baselines are not able to clearly identify the behavior to unlearn.

**Unlearning Efficiency.** Tab. 6 shows the runtime for baselines and our RFF. Though our RFF is not the most efficient, we are able to attain good unlearning quality and model utility.

$\mathcal{D}_{\text{f}}$**'s $Q$-value Distribution Post RFF.** We further provide the post-unlearning $Q$-value distribution on hopper in Fig. 8, showing that RFF does not collapse $Q$-values on $\mathcal{D}_{\text{f}}$ to minimal levels. Instead, forgotten samples settle into a lower but non-degenerated range relative to $\mathcal{D}_{\text{r}}$, confirming that RFF performs retain-calibrated suppression rather than indiscriminative value shrinking.

**Why Relative Fisher Weighting.** An ablation on Walker2D (5% unlearning) further confirms the necessity of relative Fisher weighting. We tried unlearning using the hard threshold approach in SSD (Foster et al., 2024) (denoted as SSD-CU) and the $\mathcal{D}_{\text{f}}$-Fisher weighted unlearning (denoted as $\mathcal{D}_{\text{f}}$-Fisher). SSD-CU fails to remove the designated trajectories (93.1% positive predictions), while the $\mathcal{D}_{\text{f}}$–Fisher variant improves forgetting (10.7%) but suffers notable utility loss (0.90). RFF achieves both strong forgetting (8.7%) and high utility (0.94), demonstrating that balancing forget-set and retain-set Fisher signals is essential for stable unlearning.

| Task | $\eta$ | $\beta_F$ | $\epsilon'$ | $\alpha$ | $\lambda$ | $k_\pi$ | $k_Q$ | Iters |
|------|--------|-----------|-------------|----------|-----------|---------|-------|-------|
| Cheetah-dir (Backward) | 1 | 0.95 | 2e-3 | 0.1 | 2 | 1 | 1 | 25 |
| Cheetah-dir (Forward) | 1 | 0.95 | 6e-3 | 0.1 | 2 | 1 | 1 | 75 |
| Cheetah-vel | 1 | 0.95 | 2e-3 | 0.1 | 2 | 1 | 1 | 75 |
| Ant-dir | 1 | 0.95 | 2e-3 | 0.1 | 2 | 1 | 1 | 75 |

Table 7: Hyperparameter configurations.

Figure 8: Post-RFF $Q$-value distribution on Hopper.

## 7.5 EXTENDED RELATED WORK

**Behavior Unlearning and Concept Ablation.** Behavior-level unlearning in offline RL is conceptually related to concept ablation in diffusion models, where specific semantic concepts are removed by editing model representations (Gandikota et al., 2023; Wu et al., 2024). However, the analogy is limited. Concept ablation typically operates in the latent or score space and targets high-level semantic features ("erase the concept of cars"). In contrast, behavior unlearning in RL is fundamentally data-grounded: the behavioral "concepts" correspond to structured state–action patterns supported by trajectories in the forget set. Rather than editing internal representations, RFF removes the parameter-level influence of forget set through relative Fisher weighting and retain-calibrated value suppression. When a behavior is uniquely induced by the forget set, this influence removal naturally yields behavior-level forgetting; when correlated behaviors also apprear in the retain data, strict behavioral erasure is ill-posed. Thus, RFF can be viewed as extending concept-level unlearning into the behavioral domain while remaining firmly rooted in trajectory-level data provenance.

**Machine unlearning and generative models.** Machine unlearning aims to remove the influence of training data without full retraining, often motivated by privacy regulations and the "right to be forgotten." Early strategies include partition-based retraining (*e.g.*, SISA) (Bourtoule et al., 2021), influence-function–based updates (Guo et al., 2019), and distillation to a student model trained only on retained data (Golatkar et al., 2020). More recently, unlearning has been extended to generative models, particularly diffusion and transformer architectures, where methods suppress specific concepts or examples through gradient ascent, distribution re-weighting, or score-function editing (Gandikota et al., 2023; Wu et al., 2024). While effective in supervised and generative contexts, these approaches do not address the behavioral dynamics of reinforcement learning, where forgetting must alter both policy actions and value estimates.

**Policy editing and unlearning in RL.** In reinforcement learning, undesired behaviors are often mitigated by reward penalties, action masking, or constraint learning (Garcıa & Fernández, 2015; Ray et al., 2019), but these methods impose new incentives rather than removing the influence of unwanted data. Recent work proposes actor-only unlearning techniques, such as randomizing rewards or directly penalizing the forget set (Gong et al., 2024). However, such methods cannot fully suppress the critic's value estimates, allowing forgotten behaviors to resurface. Our approach differs by jointly unlearning both the actor and the critic, while stabilizing value estimates on retain data to preserve task utility.

**Safe reinforcement learning.** Safe reinforcement learning (safe RL) focuses on preventing agents from executing unsafe or undesirable actions during training or deployment. Common approaches include reward shaping, constrained policy optimization, shielding, and action masking (Garcıa & Fernández, 2015; Ray et al., 2019). These methods proactively discourage unsafe behavior but operate prospectively, requiring constraints or safety signals to be specified in advance. In contrast, unlearning addresses a retroactive setting: the policy has already been trained on data that may later be deemed undesirable, and the goal is to remove its influence without retraining from scratch.

**Offline RL and diffusion policies.** Offline RL learns policies from static datasets, typically through conservative critics, policy constraints, or behavior regularization to handle distribution shift (Levine et al., 2020; Kumar et al., 2020). Diffusion policies extend this line by parameterizing actions as samples from a conditional denoising process (Wang et al., 2022; Chi et al., 2023; Mao et al., 2024; Fang et al., 2024), improving multimodality and robustness. However, the challenge of unlearning trajectories in diffusion-parameterized policies has not been addressed. RFF provides the first framework for unlearning in this setting, combining actor–critic suppression with retain-set stabilization to achieve both effective forgetting and performance preservation.