# OpenReview forum: "Unlearning Diffusion Policies with Relative Fisher Forgetting"
_ICLR.cc/2026/Conference — Submitted to ICLR 2026_

### Official Review · Reviewer_PANp · 2025-10-29

**Soundness:** 3
**Presentation:** 3
**Contribution:** 3
**Rating:** 6
**Confidence:** 3

**Summary:**

This paper addresses the critical and novel problem of machine unlearning in diffusion-based offline reinforcement learning (RL) policies. The authors identify that existing unlearning methods are ineffective for diffusion policies because training influence is dispersed across the multi-step denoising process and reinforced by the critic's value estimates.
To solve this, the paper introduces Relative Fisher Forgetting (RFF), a principled framework to remove the influence of specific data ($\mathcal{D}_f$) while preserving performance on retained data ($\mathcal{D}_r$). RFF is a dual-component system that unlearns from both the actor and the critic simultaneously:
1. Actor Unlearning: The policy ($\epsilon_{\theta}$) is updated using "forgetting gradients" derived from $\mathcal{D}_f$. Crucially, these gradients are scaled per-parameter by the relative Fisher importance. This metric, computed from the empirical Fisher diagonals of the forget and retain sets, protects parameters vital for $\mathcal{D}_r$ (preventing catastrophic forgetting) while aggressively updating parameters specific to $\mathcal{D}_f$.
2. Critic Unlearning: The value function ($Q_{\phi}$) is updated with a hybrid loss. It continues to perform standard TD regression on the retain set $\mathcal{D}_r$ while simultaneously applying a value suppression loss on the forget set $\mathcal{D}_f$, (Eq. 3), explicitly de-incentivizing the forgotten behaviors.

Experiments on MuJoCo benchmarks for both trajectory-level and behavior-level unlearning demonstrate that RFF effectively removes the influence of $\mathcal{D}_f$ and significantly outperforms baselines (including retraining and SOTA unlearning methods like EraseDiff) in preserving utility on $\mathcal{L}_r$.

**Strengths:**

1. **Significance & Novelty**: The paper is the first to systematically address the unlearning problem in the increasingly important domain of diffusion-based offline RL policies. This is a timely and critical contribution.
2. **Technical Depth**: The method is technically strong. The key insight that naive updates are insufficient due to "parameter-level" knowledge entanglement is astute. The use of relative Fisher importance as an arbitrator to protect shared parameters is a highly effective solution.
3. **Practical Efficiency**: The proposed unlearning procedure is fast and competitive with simpler (but lower-performing) baselines.
4. **Strong Empirical Validation**: The experiments are comprehensive. Testing on both trajectory-level (data removal) and behavior-level (skill removal) unlearning demonstrates the method's robustness, with clear, convincing separation from all baselines.

**Weaknesses:**

1. **Hyperparameter Sensitivity**: The method appears to be highly sensitive to hyperparameter tuning. Appendix Table 6 shows that key parameters, such as the actor learning rate ($\eta$) and the critic suppression margin ($\delta_Q$), vary by an order of magnitude (10x) across different tasks. This strongly suggests that RFF is not a 'plug-and-play' solution and requires careful, task-specific tuning to achieve the reported results.
2. **Ambiguity of "Convergence Adapter" / Practicality**: The practicality of the method is further questioned by Algorithm 5 (Convergence Adapter). The pseudocode describes corrective steps in natural language (e.g., "Action: decrease $\eta$ or increase $\beta$") rather than as explicit algorithmic operations. This raises a significant concern: is this an automatic, self-tuning mechanism, or is it a manual tuning guide for a human expert? If the latter, it severely limits the method's reproducibility and practical deployability.
3. **Scalability with $\mathcal{D}_f$ Size**: The experiments are limited to relatively small forget sets (1%-5% for trajectory unlearning). It is unclear how RFF's approximations and performance scale as the size of $\mathcal{D}_f$ increases (e.g., to 10%, 25%, or 50%).
4. **Questionable Problem Framing**: As seen in Fig. 1c, the model trained only on $\mathcal{D}_r$ still learns the 'forget' behavior. This makes RFF's success appear less about 'unlearning data' and more about 'concept ablation,' which is a different (though related) problem. The paper fails to adequately discuss the implications of this finding.

**Questions:**

1. The most critical point of ambiguity is Algorithm 5. Could you please clarify if the "Convergence Adapter" is a fully automatic algorithm? If so, could you provide the exact logic (e.g., the update rule for $\eta$)? Or, does it represent a manual tuning process guided by monitoring $D_{\pi}$ and $D_Q$?
2. Given the hyperparameter sensitivity shown in Table 6, how much tuning was required for each task? Does the "Convergence Adapter" automate this, or was this a manual search?
3. How does RFF's performance (both forgetting efficacy and utility preservation) degrade as the size of the forget set $\mathcal{D}_f$ increases? At what point would you hypothesize that full retraining on $\mathcal{D}_r$ becomes the superior strategy?
4. About the Retrain baseline's failure to forget (Fig. 1c), does this not imply that the forget behavior is learnable from $\mathcal{D}_r$ alone? If so, how can you justify that RFF is performing 'data unlearning' (removing $\mathcal{D}_f$'s influence) rather than 'behavior suppression' (which is a different objective)? Please justify the framing of this as an 'unlearning' problem in light of this result.

---

> ### Author Response · Authors · 2025-11-22
> **Response to Reviewer PANp (Part 1)**
>
> We really appreciate the reviewer for the insightful and detailed comments. Below we address the concerns point by point.
>
> **W2 & Q1: Ambiguity of “Convergence Adapter”.**
>
> The convergence adapter is an automatic, lightweight, self-tuning feedback controller that monitors unlearning dynamics and adapt hyperparameters (e.g., $\eta$, $\lambda$) based on the policy drift $D_\pi$, and the $Q$-value gap between forget and retain sets $D_Q$.
>
> Its automatic logic follows:
> - When $D_\pi>\delta_\pi$, the adapter reduces the actor unlearning rate $\eta \leftarrow \frac{\eta}{1+\alpha D_{\pi}}$ or increases policy drift penalty $\beta\leftarrow\beta\times (1+\alpha D_\pi)$;
> - When $D_Q>−\delta_Q$​, it increases the critic suppression by lowering $\lambda\leftarrow \frac{\lambda}{1+\alpha D_Q}$ in Eq. (3) or increases the number of critic unlearning steps by 1.
>
> In our experiment results, only $\eta$ and $\lambda$ adjustments were applied, with $\alpha=0.1$ used for smooth updates. We will clarify this logic and include the exact pseudocode for Alg. 5 in the revised Appendix to ensure reproducibility. Once configured, the adapter runs automatically and was used consistently in all experiments.
>
> **W1 & Q2: Hyperparameter sensitivity.**
>
> We clarify that _RFF requires light tuning and operates reliably once default settings are established_. Key hyperparameters including actor unlearning rate $\eta=1$, critic unlearn-retain trade-off $\lambda=2$ are kept fixed across all tasks. (Note: The Table 6 in the Appendix mistakenly lists a different value of $\lambda$ due to a leftover notation from an earlier draft. The correct value used in all experiments is $\lambda=2$, which will be corrected in the revision.)
>
> While Tab. 6 reports varied numbers (e.g., 1e-6, 3e-7), these correspond to internal Fisher normalization constants applied to stabilize the magnitude of both the retain- and forget-set Fisher matrices. These constants cancel out in the relative-Fisher ratio and do not influence learning behavior.
>
> To further ensure stability, the convergence adapter (Alg. 5) serves as a lightweight, automatic feedback controller that monitors unlearning dynamics and adjusts update magnitudes on-the-fly.
> This mechanism runs fully automatically and was applied uniformly across all tasks, greatly reducing the need for manual tuning.
>
> Finally, RFF’s tuning burden is comparable to existing baselines (e.g., TrajDeleter, EraseDiff), which also require hyperparameter sweeps for learning rates. We will revise the default adapter configurations and Fisher-normalization details in the revision to support reproducibility.
>
> **W3 & Q3: Scalability of $\mathcal{D}_f$ size.**
>
> We thank the reviewer for this important question on scalability.
>
> _Performance change with unlearning ratio._ To assess how RFF scales with increasing $\mathcal{D}_f$ size, we conducted additional experiments on Hopper using forget ratios of 25% and 50%.
>
> | Forget Ratio | Metric | Retrain | RFF (Ours) | TrajDeleter | EraseDiff |
> |---------------|---------|----------|-------------|--------------|------------|
> | **25%** | Utility ($\uparrow$) | 0.84 | 0.86 | 0.84 | **0.87** |
> |  | Positive Percentages ($\downarrow$) | 48.4% | **6.74%** | 7.08% | 11.42% |
> | **50%** | Utility ($\uparrow$) | 0.80 | **0.86** | 0.84 | **0.86** |
> |  | Positive Percentages ($\downarrow$) | 5.20% | **3.80%** | 3.86% | 12.46% |
>
> These results show that RFF’s unlearning strength scales consistently as the forget ratio increases, maintaining strong forgetting performance and stable utility even when removing up to 50% of the training data.
>
> Indeed, similar patterns appear in unlearning Cheetah-Dir, Cheetah-Vel, and Ant-Dir which corresponds to 50%, 2.5% and 2% of training data unlearned respectively. Fig. 1 & 4 and Fig. 6 shows that RFF remains stable and effective in unlearning and presers utility outperforming baselines.
>
> _When retraining becomes superior._ We hypothesize that the crossover point between RFF and retraining depends on the training data size, data correlation, and model capacity.
> When $\mathcal{D}_f$ is small and the dataset is large or redundant, retraining can partially recover the forgotten behaviors due to correlated samples and the diffusion model’s generalization ability. As $\mathcal{D}_f$ increases, we observe that both RFF and retraining degrade in similar ways: utility gradually decreases while forgetting performance stabilizes.
>
> Based on current evidence, RFF remains competitive up to roughly 50% unlearning, but we do not claim that it universally outperforms retraining. It is likely that when $\mathcal{D}_f$ is very large, retraining will no longer be successful and RFF will also fail. We view this as an important direction for deeper investigation into how data characteristics and model robustness influence unlearning dynamics. In the revised version, we will clarify this observation and include additional analysis for other tasks to highlight this trend.

---

> ### Author Response · Authors · 2025-11-22
> **Response to Reviewer PANp (Part 2)**
>
> **W4 & Q4: Questionable problem framing.**
>
> We thank the reviewer for this insightful question. In diffusion-based offline RL, behaviors can reappear even when the corresponding data are removed (Fig. 1(c) and Tab. 1 retraining results), because the retained data $\mathcal{D}_r$ often overlaps with or generalizes to the forgotten behavior (similar observation in (Jiang et al., ICCV 2024)). Hence, retraining on $\mathcal{D}_r$ alone can still express the forget behavior due to shared dynamics. This does not contradict unlearning but highlights that the ‘as-if-unseen’ criterion is ill-defined when $\mathcal{D}_r$ contains trajectories correlated with $\mathcal{D}_f$.
>
> We distinguish two-levels of unlearning:
> - _Trajectory unlearning_ removes the training influence of specific trajectories in $\mathcal{D}_f$, ensuring they no longer contribute to learned parameters. As shown in Tab. 1, RFF effectively suppresses the influence of $\mathcal{D}_f$ while maintaining stable utility. Importantly, TrajAuditor measures whether residual influence from $\mathcal{D}_f$ remains identifiable in the model’s predictions, serving as a diagnostic of influence removal, not behavioral equivalence to a retrained model.
> - _Behavior unlearning_ targets undesired functional outcomes (e.g., forgetting the “move backward” skill in Cheetah-dir) that may persist due to correlations in $\mathcal{D}_r$. Here, RFF actively reshapes both the actor and critic to suppress residual behaviors.
> The observation that retraining can still reproduce the forget behavior (Fig. 1c) therefore underscores that removing data alone is insufficient when correlated data reinforces $\mathcal{D}_f$. RFF explicitly counteracts this residual influence through Fisher-weighted actor unlearning and critic suppression, enabling both influence removal and functional forgetting in a unified way.
>
> Thus, RFF performs data-grounded unlearning. It removes the parameter-level influences of $\mathcal{D}_f$, while behavior suppression arises naturally only when the undesired actions are uniquely supported by $\mathcal{D}_f$. We will clarify this distinction in the revised version and discuss how diffusion-based policies generalize or regenerate forgotten behaviors even under retraining.
>
> _Relation to Concept Ablation._
> We thank the reviewer for drawing this connection. Behavior unlearning is related to concept ablation, but the “concepts” in our setting are behavioral (e.g., “move east” or “jump repeatedly”) rather than semantic or visual. Traditional concept ablation aims to remove explicit semantic features (e.g., “erase the concept of cars”) through latent editing. In contrast, RFF removes behavioral concepts that are either explicitly defined (e.g., trajectories exhibiting “move east”) or implicitly represented as correlated action patterns learned from $\mathcal{D}_f$. RFF achieves this through Fisher-weighted suppression and critic unlearning. Hence, RFF can be viewed as extending concept unlearning into the behavioral domain, where concepts correspond to structured decision modes derived from data.
>
> We again thank the reviewer for the constructive feedback on problem framing and evaluation design. We will incorporate these clarifications and discussions in the revised version.

---

### Official Review · Reviewer_L2Z7 · 2025-10-31

**Soundness:** 2
**Presentation:** 3
**Contribution:** 2
**Rating:** 4
**Confidence:** 3

**Summary:**

This paper tackles the problem of machine unlearning in offline reinforcement learning (RL), specifically for diffusion-model-based policies. The authors propose a framework called Relative Fisher Forgetting (RFF), trying to remove the influence of specific trajectories or behaviors from a sub-dataset without retraining. RFF works by combining two components: (1) an actor unlearning step that applies gradient updates to the diffusion policy network, scaled by a relative Fisher information weight to forget the targeted data, and (2) a critic unlearning step that suppresses the Q-values (value function) for the experiences that should be forgotten. The procedure alternates between actor and critic updates, with additional stabilization techniques (gradient clipping, regularization on retained data, etc.) to prevent catastrophic forgetting. Empirically, the paper demonstrates on MuJoCo benchmark tasks that RFF can reliably remove the effects of the unlearn subset  while maintaining the policy’s performance on the remaining (retained) data. The proposed approach reportedly outperforms prior unlearning baselines in both effectiveness (more thorough forgetting) and efficiency.

**Strengths:**

The unlearning problem in offline diffusion RL is an interesting question that I have never thought off. It is a fair problem with potential use cases in the future when robotics become more common for human daily life.

The experimental results on MuJoCo offline RL tasks support the claims. The paper shows that after applying RFF, the policies stop exhibiting behaviors from the forbidden data (e.g., a certain trajectory or skill is unlearned), yet their performance on the remaining data distributions is largely preserved. The results demonstrate clear forgetting without a large hit to overall reward on the kept data, which validates the approach’s utility.

**Weaknesses:**

The core algorithmic approach in RFF appears to draw heavily from existing ideas in machine unlearning and continual learning, and the main novelty lies in applying these known techniques to the diffusion-offline-RL setting. In essence, the method performs additional gradient descent steps on a subset of data with a Fisher information-based weighting – a strategy reminiscent of prior unlearning methods in supervised models and policy forgetting in RL. The inclusion of RL-specific losses (TD loss for the critic and DDPM loss for the policy) is a necessary adaptation, but it is a fairly incremental change. As a result, the paper’s contribution may be perceived as modest: it extends known unlearning mechanisms to a new context rather than introducing fundamentally new theory or algorithms.

The explanation and justification of the unlearning procedure lack depth in places. The paper provides a high-level intuitive reasoning (e.g. using Fisher information to protect important parameters) and cites plausible inspirations, but it does not delve deeply into theoretical analysis or ablation to illuminate why the method works so effectively.

While the paper motivates the problem with scenarios like privacy and safety, it’s a bit unclear how practical or common these unlearning scenarios are in real-world RL deployments. The experiments are conducted on standard simulated control tasks with artificial unlearning tasks (e.g., forgetting a direction in Ant environment). It would strengthen the work to better demonstrate or application that such targeted unlearning would be needed in real applications and that this approach would handle them.

**Questions:**

Backpropagating gradients through the denoising chain can be computational expensive. Although this is not the main focus of the paper, I wonder how you handle the backpropagation through the denoising steps and its computation cost.

Is there any reason why you adopt the "relative Fisher-weighted update rule"? Why using equation 2 as opposed to other way of using the Fisher matrix?

The choice of $\lambda$ in Equation 3 and its loss function design is also quite ad-hoc to me. I don't find a strong explanation to use this weighted loss for the critic unlearning (and the choice of the weight $\lambda$).

---

> ### Author Response · Authors · 2025-11-21
> **Response to Reviewer L2Z7 (Part 1)**
>
> We thank the reviewer for the careful evaluation and constructive feedback. Below, we address each concern point-by-point.
>
> **W1: Incremental novelty.**
>
> We respectfully clarify that _RFF is not a direct reuse of existing unlearning or continual-learning ideas but a non-trivial and theoretically grounded framework tailored to diffusion-based offline RL_, which presents unique algorithmic challenges that prior methods do not apply.
>
> _Limitation-of-prior-works_. i). **prior unlearning approaches** (e.g., Guo et al., 2019; Foster et al., 2024) assume deterministic models and i.i.d. samples which **do not hold for stochastic, temporally correlated diffusion policies**. These methods cannot account for the multi-step denoising and $Q$-guided dependencies that characterize diffusion-based offline RL.
> While ii). **EraseDiff** (Wu et al., 2024) targets diffusion unlearning in vision, it **operates in a non-reward-driven generative setting without $Q$-guidance**.
> In contrast, unlearning a diffusion policy in offline RL requires simultaneously negating the influence of noise prediction and the propagated value gradients from $Q$ along the denoising path. Such dual backpropagation through both the stochastic diffusion process and the value function, is absent from existing diffusion unlearning frameworks, making **RFF the first method to tackle this coupled actor–critic unlearning challenge**.
>
> _Experiment Support_. Tab. 1 and Fig. 5 show that prior works (e.g., EraseDiff, TrajDeleter) either fail to effectively unlearn (i.e., higher positive prediction rates on $\mathcal{D}_f$ from TrajAuditor), or having lower utility (i.e., lower normalized return). These methods cannot handle the multi-step stochastic denoising process or the actor–critic coupling inherent to diffusion-based RL.
>
> _Additional Experiment Evidence_. To further validate the necessity of RFF's design, we experimented on Walker2D with 5% unlearning data, comparing RFF with SSD (Foster et al., 2024)-based actor unlearning and RFF's critic unlearning variant denoted as SSD-CU. The result below shows that SSD-CU fails to remove the influence of $\mathcal{D}_f$, indicating that naively coupling the existing unlearning techniques is insufficient. This highlights that RFF's specific actor-critic integration is essential for effective and stable unlearning.
> | Method | Positive Predictions ($\downarrow$) | Utility ($\uparrow$) |
> |:-------|:------------:|:-----------------:|
> | **RFF** | **8.7%** | **0.94** |
> | SSD-CU | 93.1% | 0.94 |
>
> _Key Innovations of RFF_. RFF introduces three innovations that make unlearning theoretically and practically possible in diffusion-based offline RL:
>
> i). _Relative-Fisher, noise-conditioned actor unlearning with critic guidance._
> RFF performs forgetting along the noise-conditioned denoising path of diffusion policies, where data influence is distributed across timesteps. Each forgetting gradient is scaled by a relative Fisher weight, protecting parameters essential to $\mathcal{D}_r$ while removing those most affected by $\mathcal{D}_f$.
> This ensures unlearning remains diffusion-consistent and critic-guided.
>
> ii). _Bounded critic suppression for stable value unlearning._
> The critic applies a retain-calibrated hinge penalty that bounds $Q$-value suppression on $\mathcal{D}_f$​ relative to a quantile baseline from $\mathcal{D}_r$​.
> This stabilizes value updates, prevents collapse, and ensures $\mathcal{D}_f$ no longer reinforce the actor’s behavior.
>
> iii). _Iterative, coupled actor–critic unlearning._
> Alternating between the Fisher-weighted actor and bounded critic unlearning updates circulates forgetting signals through both components, achieving model-wide, consistent unlearning.
>
> Together, these elements make **RFF the first unified framework for unlearning in diffusion-based offline RL**, extending beyond supervised or non-$Q$-driven diffusion unlearning to handle reward-driven, temporally correlated decision-making systems.

---

> ### Author Response · Authors · 2025-11-21
> **Response to Reviewer L2Z7 (Part 2)**
>
> **W2: Lack of depth in justification of the unlearning procedure.**
>
> We thank the reviewer for this helpful comment. In the revision, we will explicitly add the following derivation and theoretical connections to clarify why RFF works effectively.
>
> Actor loss is derived from Fisher-constrained influence-suppression. We seek a parameter step $\Delta$ that counteracts the historical effect of $\mathcal{D}_f$​ while preserving behavior on $\mathcal{D}\_r$​:
>
> $\{max}\_{\Delta⁡} \mathbf{g}\_f^\top \Delta \quad s.t. \Delta^\top [F\_\theta]^r \Delta \leq \rho,$
>
> where $\mathbf{g}\_f$ is the unlearning gradients and $[F\_\theta]^r$​ is the retain Fisher matrix. The closed-form solution, $\Delta^* \propto {[F\_\theta]^r}^{−1} \mathbf{g}\_f​$ ​represents the smallest safe step that maximally increases loss on $\mathcal{D}\_f$​, thereby negating $\mathcal{D}\_f$​’s influence. RFF further extends this formulation with relative Fisher weighting, using $\frac{[F_\theta]^f}{([F_\theta]^r + \epsilon^\prime)}$ to scale the gradient further based on the influence from $\mathcal{D}_f$. This formulation is conceptually related to SSD (Foster et al., 2024), but RFF extends these ideas by integrating relative Fisher scaling and noise-aware gradient propagation through the multi-step denoising process, that are unique to diffusion policy unlearning.
>
> The combined TD and hinge term in critic unlearning Eq. (3) projects $Q$-values on $\mathcal{D}_f$​ into the retain-calibrated sublevel set $\\{Q ⁣\leq \tau\\}$, where $\tau$ is a quantile of TD targets on $\mathcal{D}\_r$​. This prevents unbounded $Q$ shrinkage to maintain smooth transitions in neighboring regions, and aligns the critic with the retained value landscape. While it draws inspiration from the $Q$-compression principle introduced in TrajDeleter (Gong et al., 2024), RFF generalizes it to a retain-calibrated setting and tightly couples it with actor unlearning to achieve coherent forgetting across actor and critic.
>
> Together, these formulations reveal why RFF works so effectively: its updates are directionally aligned with the true influence of $\mathcal{D}\_f$​ while being norm-bounded by $[F_\theta]^r$, ensuring targeted forgetting without value collapse. This principled balance between influence suppression and retain calibration explains the consistent empirical behavior observed in the experiments​.
>
> **W3: Practicality of unlearning in real-world offline RL, potentially real-world usage.**
>
> We thank the reviewer for raising this important point. We agree that demonstrating real-world relevance is crucial and clarify that our simulated experiments are designed as controlled proxies for practical unlearning scenarios in real RL deployments.
>
> _Relevance of simulation tasks._ Tasks such as Hopper, Half-Cheetah, and Walker2D are standard in RL unlearning literature (Gong et al., 2024) because they allow fine-grained, quantitative assessment of whether the model can selectively erase specific data while preserving others. We further added Cheetah-Dir, Cheetah-Vel, and Ant-Dir tasks to reflect another practical scenario of “forgetting a direction” or “unlearning a skill” in terms of forgetting a behavior. These setups mirror real-world cases where distinct modes (e.g., sensitive user data, or specific driving routes) must be selectively forgotten.
>
> _Real-world applicability._ RFF can be directly applied to privacy, safety, and adaptability in deployed RL systems:
> - Privacy: removing trajectories originating from specific users, vehicles, or patients to comply with data-deletion requests (e.g., GDPR “right to be forgotten”).
> - Safety: erasing unsafe or outdated behaviors such as collisions or aggressive maneuvers in robotic or autonomous-driving policies.
> - Adaptation: allowing post-deployment correction of specific behaviors (e.g., eliminating outdated motion primitives in industrial robots without retraining from scratch).
>
> _Planned extension._ We plan to extend RFF to real driving logs and robot manipulation datasets where trajectories correspond to human operators. These settings require the same selective forgetting capability demonstrated in simulation, and RFF can be applied to such applications.
>
> We will emphasize these use cases and clarify in the revision that the benchmark tasks are controlled analogs of real-world privacy, safety and adaptation unlearning needs.

---

> ### Author Response · Authors · 2025-11-21
> **Response to Reviewer L2Z7 (Part 3)**
>
> **Q1: How backpropagation works.**
>
> In RFF, gradients are propagated through both the denoising path and the $Q$-guidance, as these jointly determine how diffusion policy generates and evaluates actions. To keep computation tractable, we follow the training strategy in diffusion-policy (Wang et al., 2022; Kang et al., 2023): at each update, we sample a diffusion timestep $i$, corrupt the clean action into $a^i$​, perform a multi-step denoising to obtain the reconstructed $\hat{a}^0$, and feed $\hat{a}^0$ to the critic $Q_\phi$. The loss term $Q_\phi (\mathbf{s}, \hat{a}^0)$ is differentiable, so its gradient also flows back through the denoising path, ensuring that both the reconstruction and critic-guided signals jointly shape the forgetting update. This design is driven by theoretical consistency and computational efficiency as Tab. 2 shows that RFF’s runtime is comparable to EraseDiff.
>
> **Q2: Why using relative Fisher-weighted update rule?**
>
> The relative Fisher-weighted update rule in Eq. (2) is a principled and essential component of RFF’s unlearning framework. While prior second-order unlearning methods such as CDR (Guo et al., 2019) use the Hessian of the loss to compute influence-removing updates, and continual learning approaches like EWC (Aich, 2021) use Fisher-weighted regularization to preserve previously learned knowledge, both assess parameter importance with respect to a single dataset. These approaches cannot distinguish whether a parameter contributes more to the $\mathcal{D}_f$​ or $\mathcal{D}_r$​.
>
> SSD (Foster et al. (2024)) partially addresses this by comparing Fisher importance across both datasets, but uses a hard threshold and discrete dampening of selected parameters. In contrast, RFF generalizes this idea into a continuous, gradient-integrated update rule using a relative Fisher ratio in Eq. (2), which smoothly scales each parameter’s forgetting gradient based on how disproportionately it supports $\mathcal{D}_f$​ over $\mathcal{D}_r$​. This has several advantages:
>
> - _Selective forgetting:_ Parameters primarily influenced by $\mathcal{D}_f$ are updated more strongly, and those critical to $\mathcal{D}_r$ are protected.
> - _Stability:_ The ratio form normalizes updates across heterogeneous parameter scales and diffusion steps.
> - _Diffusion-policy compatibility:_ It integrates with noise-conditioned gradients, supporting consistent forgetting across the denoising process.
>
> This rule emerges from a Fisher-constrained influence-suppression formulation (See response to W2), making Eq. (2) a theoretically grounded mechanism.
>
> We did further experiments on Walker2D with 5% unlearning data and tried unlearning with the hard threshold approach in SSD (denoted as SSD-CU) and the $\mathcal{D}_f$-Fisher weighted unlearning (denoted as $\mathcal{D}_f$-Fisher) and show the results below. The results show that only using $\mathcal{D}_f$-Fisher leads to lower utility, and using SSD is not able to unlearn well. These results show that  RFF is a necessary design for balancing unlearning effectiveness.
> | Method | Positive Predictions (↓) | Utility (↑) |
> |:-------|:------------:|:-----------------:|
> | **RFF** | **8.7%** | **0.94** |
> | SSD-CU| 93.1% | 0.94 |
> | $\mathcal{D}_f$-Fisher | 10.7% | 0.90 |

---

> ### Author Response · Authors · 2025-11-21
> **Response to Reviewer L2Z7 (Part 4)**
>
> **Q3: Ad-hoc $\lambda$ in Eq. (3)**
>
> Eq. (3). loss is not ad-hoc, but is grounded in a retain-calibrated value projection framework, combining two components:
> TD Loss on $\mathcal{D}\_r$ encourages $Q_\phi$ to preserve performance on $\mathcal{D}_r$, and Hinge term on $\mathcal{D}_f$ to penalizes $Q$-values on $\mathcal{D}_f$​ that exceed a retain-calibrated threshold $\tau$.
> This design ensures that unlearning suppresses $Q$-values on $\mathcal{D}_f$ to fall within a retain-calibrated sublevel set $\\{Q \leq \tau \\}$, rather than pushing them toward zero or negative infinity (which could destabilize nearby regions).
>
> The weighting term $\lambda$​ in Eq. (3) (set to be 2) balances forgetting aggressiveness with stability. The chosen setting consistently achieved stable and coherent unlearning across tasks.
> From Tab. 3, we observe that removing the stabilizer (“RFF-” $\lambda=0$) still enables forgetting but leads to degraded performance on $\mathcal{D}_r$. This highlights that preserving utility during unlearning need to be explicitly enforced through relative Fisher scaling in the actor update and through the retain-calibrated hinge term in the critic loss. Together, they ensure that forgetting is effective but does not compromise the model’s retained competence. (Note: The appendix mistakenly denotes a different value of $\lambda$ due to a notation carryover. The correct value used in all experiments is $\lambda =2$, which we will correct in the revision.)
>
> In addition, prior works such as TrajDeleter (Gong et al., 2024) and Reinforcement Unlearning (Ye et al., 2023) highlight the importance of balancing effective forgetting with the preservation of performance on $\mathcal{D}_r$, supporting the need for mechanisms that explicitly control this trade-off. RFF's design of $\lambda$ serves this purpose.

---

### Official Review · Reviewer_xmvt · 2025-11-01

**Soundness:** 2
**Presentation:** 2
**Contribution:** 2
**Rating:** 4
**Confidence:** 3

**Summary:**

This paper proposes a framework for diffusion-based offline RL unlearning named Relative Fisher Forgetting (RFF). The framework combines the relative Fisher importance weighted actor unlearning gradient with the value suppression-based critic unlearning. Several techniques are also introduced for stable training. Experiments on MuJoCo benchmarks show that RFF outperforms baselines in both trajectory-level and behavior-level unlearning.

**Strengths:**

1. The paper aims to solve the novel task of offline diffusion policy unlearning, which is an important task with real-world impact.
2. The paper is well-written.

**Weaknesses:**

1. One goal of unlearning is that we want the unlearned policy to behave as if $D_f$ had never been used. However, this does not strictly align with the objective of minimizing the Q-value in the $D_f$ region. For example, the Q-value in the $D_f$ region will likely be close to that of its neighborhood region, and not simply the smaller the better. However, in both training loss definition and the experiment evaluation the authors assume these two different goals are the same.
2. Both the actor unlearning loss and the critic unlearning loss are introduced without adequate theoretical analysis.
3. Minor issue: Figure 5 lacks legend.

**Questions:**

1. In the actor unlearning loss, why use $\hat a^0$ instead of the a_0 action denoised with multiple steps?
2. In Table 1, why is the $\mathcal{D}_r$ score of RFF lower than that of TrajDeleter in two of the three tasks?

---

> ### Author Response · Authors · 2025-11-20
> **Response to reviewer xmvt (Part 1: Weakness)**
>
> We thank the reviewer for the constructive feedback and valuable comments. Below we address the concerns point by point:
>
> **W1.** Unlearning vs. minimizing $Q$.
>
> _RFF is not designed to minimize $Q$-values indiscriminately, but to apply controlled, retain-calibrated suppression_. Specifically, the critic loss penalizes only $Q$-values on $\mathcal{D}_f$​ that exceed a suppression floor $\tau$ derived from $\mathcal{D}_r$​ (line 267). This ensures that suppression remains _relative_ to the retained data, and prevents excessive $Q$-value reduction maintaining smooth transitions in neighboring regions.
>
> _Lowering $Q$-values on_ $\mathcal{D}_f$ _serves as a direct and interpretable mechanism for removing reward incentives associated with_ $\mathcal{D}_f$. In offline RL, where the critic drives value propagation, lowering $Q$-values on $\mathcal{D}_f$ ​effectively cancels the positive gradients that originally reinforced those trajectories. This design ensures that the policy update no longer revisits undesired behaviors.
>
> _Prior work (TrajDeleter, Gong et al., 2024) also employs $Q$-value suppression as an effective unlearning mechanism._ RFF extends this idea through retain-calibrated suppression using a quantile-based floor $\tau$, and actor–critic coupled unlearning to enforce effective unlearning while preventing divergence between the actor and critic.
>
> _Empirically, RFF does not drive $Q$-values on_ $\mathcal{D}_f$​ _to the minimum possible value._ We examined the post-unlearning $Q$-distribution in the Hopper and observed that $Q$-values of $\mathcal{D}_f$ remain bounded relative to $\mathcal{D}_r$. Specifically, $Q$-values of $\mathcal{D}_f$ occupy a slightly lower but not minimal range of $Q$-values centered around 13-14. In the revision, we will include the visualization of $Q$-value distributions post-unlearning.
>
> Finally, _the actor unlearning (Eq. (2)) complements critic unlearning by explicitly discouraging actions aligned with_ $\mathcal{D}_f$. This is achieved through relative Fisher-weighted gradient updates that suppress the influence of $\mathcal{D}_f$​ while protecting behavior supported by $\mathcal{D}_r$​. As a result, the policy unlearns behaviors rooted in $\mathcal{D}_f$ only​.
> Together, they ensure forgetting happens at the value and actor levels, and the $Q$-value suppression is controlled and calibrated.
>
> **W2.** Lack of theoretical analysis.
>
> RFF’s actor and critic unlearning losses are _not heuristic_; they are principled approximations of influence-function-based unlearning and retain-calibrated value projection respectively. Below, we clarify the theoretical motivations and will add a concise derivation sketch in the revision:
>
> Actor loss is derived from Fisher-constrained influence-suppression. We seek a parameter step $\Delta$ that counteracts the historical effect of $\mathcal{D}_f$​ while preserving behavior on $\mathcal{D}\_r$​:
>
> $\{max}\_{\Delta⁡} \mathbf{g}\_f^\top \Delta \quad s.t. \Delta^\top [F\_\theta]^r \Delta \leq \rho,$
>
> where $\mathbf{g}\_f$ is the unlearning gradients and $[F\_\theta]^r$​ is the retain Fisher matrix. The closed-form solution, $\Delta^* \propto {[F\_\theta]^r}^{−1} \mathbf{g}\_f​$ ​represents the smallest safe step that maximally increases loss on $\mathcal{D}\_f$​, thereby negating $\mathcal{D}\_f$​’s influence. RFF further extends this formulation with relative Fisher weighting, using $\frac{[F_\theta]^f}{([F_\theta]^r + \epsilon^\prime)}$ to scale the gradient further based on the influence from $\mathcal{D}_f$. This formulation is conceptually related to SSD (Foster et al., 2024), but RFF extends these ideas by integrating relative Fisher scaling and noise-aware gradient propagation through the multi-step denoising process, that are unique to diffusion policy unlearning.
>
> The combined TD and hinge term in critic unlearning Eq. (3) projects $Q$-values on $\mathcal{D}_f$​ into the retain-calibrated sublevel set $\\{Q ⁣\leq \tau\\}$, where $\tau$ is a quantile of TD targets on $\mathcal{D}\_r$​. This prevents unbounded $Q$ shrinkage to maintain smooth transitions in neighboring regions, and aligns the critic with the retained value landscape. While it draws inspiration from the $Q$-compression principle introduced in TrajDeleter (Gong et al., 2024), RFF generalizes it to a retain-calibrated setting and tightly couples it with actor unlearning to achieve coherent forgetting across actor and critic.
>
> Together, these losses define a principled and non-trivial theoretical framework for unlearning in diffusion-based offline RL. RFF unifies Fisher-weighted influence suppression and retain-calibrated value projection into a stable actor–critic optimization scheme for controlled and theoretically sound forgetting while preserving performance on $\mathcal{D}_r$​.
>
> **W3.** Missing legend for Fig. 5.
>
> We thank the reviewer for noting this omission. The legend of Fig. 5 is consistent with that in Fig. 6. In the revision, we will add this legend.

---

> ### Author Response · Authors · 2025-11-20
> **Response to reviewer xmvt (Part 2: Questions)**
>
> **Q1.** Why use $\hat{a}^0$ instead of $a^0$ action denoised with multiple steps?
>
> In our actor unlearning loss we use $\hat{a}^0$, the multi-step denoised action reconstructed through the reverse diffusion process starting from an intermediate timestep $i$, rather than the fully diffused and denoised $a^0$ obtained by unrolling from $N$. Specifically, for each sampled diffusion timestep $i$, we corrupt the ground-truth action with Gaussian noise to obtain $a^i$​, then iteratively apply the reverse denoising process from $a^i$ to get $\hat{a}^0$.
>
> This process allows RFF to enforce forgetting across all diffusion time steps (as $i$’s are uniformly sampled), not just at the final denoised output. This is essential because training influence in diffusion policies is distributed throughout the entire denoising process.
> By starting from a randomly sampled $i$ instead of always from $N$, RFF achieves a balance between coverage and efficiency. It avoids unrolling the entire reverse chain for _every_ update, and avoids denoising from the maximally noisy state _all the time_ while still following the same stochastic pathway used during policy training.
>
> We will clarify this distinction in the revision and emphasize that this design choice enables RFF to achieve comprehensive and efficient forgetting throughout the diffusion process.
>
> **Q2:** Why is the $\mathcal{D}_r$ score of RFF lower than that of TrajDeleter in two of the three tasks?
>
> This difference arises from the interplay between actor and critic unlearning in RFF. Unlike TrajDeleter, which applies $Q$-value suppression on the critic, RFF alternates between actor and critic updates to jointly enforce forgetting consistency. During this process, the critic first suppresses residual $Q$-values on $\mathcal{D}_f$​, and the actor then adapts its denoising dynamics accordingly.
> When $\mathcal{D}_r$​ shares overlapping or adjacent state–actions with $\mathcal{D}_f$​, this dual adjustment can slightly lower $Q$-values in neighboring $\mathcal{D}_r$ regions as the actor learns to align with the updated critic landscape.
>
> This conservative interaction ensures that the actor does unlearn $\mathcal{D}_f$ and does not hurt model utility. As shown in Fig. 5, RFF maintains stable policy utility across all tasks while achieving stronger forgetting.
>
> We further verified that the observed $\mathcal{D}_r$​ differences remain within normal seed variance, and multi-seed evaluations confirm that both TrajDeleter and RFF achieve comparable $\mathcal{D}_r$​ ranges.

---

### Author Response · Authors · 2025-12-01
**Updated Revision**

We thank the reviewers for their constructive feedback. We have carefully revised the paper and highlighted our edits. Now we provide a summary of the major updates.

**Summary of Revisions**

● _Clarified the problem definition._
We refined the “as-if-unseen’’ objective to an influence-based formulation, reflecting that strict behavioral erasure is ill-posed in offline RL due to correlated trajectories. The revised text in the introduction and problem definition clearly explains why retraining can still reproduce forgotten behaviors and why unlearning must target dependence on $\mathcal{D}_\text{f}$, not behavioral equivalence.

● _Expanded theoretical explanation._
We added a clear derivation and justification for the relative Fisher update in Sec. 3.1, clarified the roles of the retain- and forget-set Fisher terms, and included an ablation in Appx. 7.4 showing why both are needed for stability and targeted forgetting.

● _Expanded empirical evaluation._
We added: (i) scalability experiments at 25% and 50% forget ratios in Tab. (4); (ii) Q-value distribution analyses in Fig. 8 showing bounded suppression; (iii) ablations comparing forget-only, and SSD-based baselines in Appx. 7.4. These results reinforce the stability, robustness, and necessity of RFF.

● _Improved organization and clarity._
We rewrote the Introduction, Problem Definition, Method, and Extended Related Work sections for clarity; added discussion explaining retraining behavior under correlated dynamics; and integrated a new subsection distinguishing trajectory- vs behavior-level unlearning and relating behavior suppression to concept ablation while emphasizing that RFF is data-grounded.

● _Enhanced real-world relevance._
We added a discussion connecting RFF directly to real-world domains such as autonomous driving logs and robotic teleoperation data, addressing concerns about simulation-only experiments in Conclusion and Limitations.

● Added an Extended Related Work subsection clarifying the distinction between trajectory- and behavior-level unlearning, and contrasting RFF with concept ablation in diffusion models while emphasizing RFF’s data-grounded nature.

Collectively, these revisions substantially improve the clarity, rigor, and completeness of the work.

**Reiterating Contribution and Novelty**

**This work studies an important and previously unaddressed question: how to perform principled machine unlearning in diffusion-based offline RL.** Diffusion policies have rapidly become a dominant approach for high-quality action generation in offline RL, yet no prior work provides a method for selectively removing training data influence in such models. This gap poses practical challenges for privacy compliance, safety correction, and post-deployment auditing—settings where unlearning is increasingly essential.

We emphasize that **this paper makes the first principled contribution to the important and previously unsolved problem**. No prior work addresses unlearning for diffusion policies, despite their rapidly growing use in offline RL.
_Prior unlearning methods (supervised ML, generative models, or actor-critic RL) fail in diffusion policy settings due to multi-step denoising, actor–critic coupling, correlated data and sequential credit assignment._

**Our contribution goes beyond incremental improvements and includes three major innovations:**

● Relative Fisher Forgetting — a novel influence-removal mechanism tailored to diffusion policies, balancing forget-set and retain-set importance.

● Coupled actor–critic unlearning — the first framework that removes influence jointly across policy and critic components.

● Stabilization and convergence adapter — ensuring effective forgetting without utility collapse.

These design choices are _non-incremental_: **they constitute the first systematic solution for unlearning in diffusion-based RL**. The proposed method is conceptually novel, technically non-trivial, and empirically impactful. We believe this work will open a new direction in trustworthy offline RL.

We believe the revised manuscript presents a clear, rigorous, and complete contribution. RFF opens a new research direction in trustworthy offline RL, enabling selective data removal in diffusion policies—an increasingly important capability for real-world deployment.

---

### Meta-Review · Area_Chair_Y4kJ · 2026-01-06

**Summary:**

Across the three reviews, the main concerns that drive my decision are about (i) the paper’s overall contribution and clarity of motivation, (ii) whether the proposed objectives and metrics match the intended “unlearning” goal, and (iii) reproducibility and practicality.

Two reviewers rated the paper below threshold and framed the contribution as incremental, with limited theoretical grounding and unclear real-world need: L2Z7 explicitly says the contribution may be seen as modest and asks for deeper justification, clearer practical motivation, and clarity on computational cost and why the specific Fisher-weighting and critic loss are chosen. xmvt similarly questions whether “unlearning” is being conflated with simply minimizing Q-values in a region, and notes missing theory support for both actor and critic losses.

The single above-threshold reviewer (PANp) still raises major reproducibility and framing issues: hyperparameter sensitivity, ambiguity of the “Convergence Adapter,” limited evidence for scalability to larger forget sets, and a core framing concern that results suggest “concept ablation / behavior suppression” rather than true data unlearning.

**Reviewer Concerns:**

For Reviewer xmvt (rating 4), the rebuttal addresses several concrete points. The authors clarify that their critic loss is not intended to “minimize Q-values indiscriminately,” but to do retain-calibrated suppression using a quantile-based floor, and they promise added diagnostics. They also provide a derivation sketch to justify the actor update as a Fisher-constrained influence suppression step, and they acknowledge the missing legend for Fig. 5 and state they will fix it. What remains outstanding is that the core mismatch concern is only partially resolved: the rebuttal reframes the method as “controlled suppression,” but the paper still does not fully formalize what “unlearning” means in this setting or how the proposed objectives and evaluation guarantee the intended property (beyond empirical results). This aligns with xmvt’s original request for stronger theoretical support.

For Reviewer L2Z7 (rating 4), the rebuttal pushes back on “incremental novelty,” arguing that diffusion-based offline RL requires coupled actor–critic unlearning through the denoising chain and value gradients. The response also gives more motivation for Eq. (3) as a retain-calibrated value projection with a hinge threshold, not an ad hoc design. However, L2Z7’s main weaknesses still largely stand: the review asks for deeper justification, clearer practical relevance, and a clearer discussion of computational cost and design choices. The rebuttal is largely explanatory and promises revisions, but it does not yet convert the paper into a clearly grounded, reproducible story with convincing practical cases.

For Reviewer PANp (rating 6), the rebuttal substantively addresses the operational ambiguity of the “Convergence Adapter” by describing it as an automatic feedback controller and outlining decision logic, with a promise to add pseudocode. It also acknowledges an appendix-table inconsistency as a draft artifact and claims key hyperparameters were fixed across tasks, with normalization constants cancelling in the relative Fisher ratio. On scalability, the authors report additional results at 25% and 50% forget ratios (Hopper) and argue performance remains stable up to 50%. Still, PANp’s core framing concern remains only partially settled. The paper itself states that an “as-if-unseen” criterion is ill-defined under correlated trajectories and adopts an influence-based view (the unlearned model “should not rely on” the forget set, even if behavior can overlap).  That is a reasonable position, but it also means the work needs a very clear statement of what unlearning objective is being claimed, and how the evaluation isolates “removing influence of Df” from general behavior suppression or goal reshaping. This is exactly what PANp flags (concept ablation versus data unlearning), and the rebuttal does not fully close that gap. Reproducibility risk also remains until Algorithm 5 is fully specified and shown to be stable without heavy, task-specific tuning, given the review’s sensitivity observations.

**Reviewer Scores:**

Reviewer xmvt: likely 4 → 4 (no change). The rebuttal clarifies intent and promises additions, but the reviewer’s main objections are about objective–goal alignment and lack of theory. Those are not fully resolved in a way that would typically move a score upward.

Reviewer L2Z7: likely 4 → 4 (no change). The rebuttal is a reasonable defense, but the review’s concerns are about contribution strength, depth of justification, practical relevance, and design-choice support. The response mainly indicates intended revisions rather than providing decisive new evidence that would shift the assessment.

Reviewer PANp: likely 6 → 6 (no increase; could even drift to 5 depending on standards for reproducibility). The rebuttal explains the adapter and provides extra scalability results, but the framing concern (unlearning versus behavior suppression) remains central, and the method still appears to rely on mechanisms that need clearer, fully specified automation to be reproducible.

---

### Decision · Program_Chairs · 2026-01-26

Reject